# Synchrony-Gated Plasticity with Dopamine Modulation for Spiking Neural Networks

**Yuchen Tian**[*]                                                                                           *yuchen.tian@sydney.edu.au*
*School of Biomedical Engineering, The University of Sydney, Sydney, NSW, Australia*

**Samuel Tensingh**                                                                                       *samuel.tensingh@sydney.edu.au*
*School of Biomedical Engineering, The University of Sydney, Sydney, NSW, Australia*

**Jason K. Eshraghian**                                                                                                   *jsn@ucsc.edu*
*Dept. of Electrical and Computer Engineering, University of California, Santa Cruz, CA, USA CIFAR Fellow*

**Nhan Duy Truong**                                                                                       *nhan@brainconnect.com.au*
*School of Biomedical Engineering, The University of Sydney, Sydney, NSW, Australia*

**Omid Kavehei**                                                                                           *omid.kavehei@sydney.edu.au*
*School of Biomedical Engineering, The University of Sydney, Sydney, NSW, Australia*

**Reviewed on OpenReview:** *https://openreview.net/forum?id=Gx4Qk6NtEP*

## Abstract

While surrogate backpropagation proves useful for training deep spiking neural networks (SNNs), incorporating biologically inspired local signals on a large scale remains challenging. This difficulty stems primarily from the high memory demands of maintaining accurate spike-timing logs and the potential for purely local plasticity adjustments to clash with the supervised learning goal. To effectively leverage local signals derived from spiking neuron dynamics, we introduce Dopamine-Modulated Spike-Synchrony-Dependent Plasticity (DA-SSDP), a synchrony-based rule that is sensitive to loss and brings a synchrony-based local learning signal to the model. DA-SSDP condenses spike patterns into a synchrony metric at the batch level. An initial brief warm-up phase assesses its relationship to the task loss and sets a fixed gate that subsequently adjusts the local update's magnitude. In cases where synchrony proves unrelated to the task, the gate settles at one, simplifying DA-SSDP to a basic two-factor synchrony mechanism that delivers minor weight adjustments driven by concurrent spike firing and a Gaussian latency function. These small weight updates are only added to the network's deeper layers following the backpropagation phase, and our tests showed this simplified version did not degrade performance and sometimes gave a small accuracy boost, serving as a regularizer during training. The rule stores only binary spike indicators and first-spike latencies with a Gaussian kernel. Without altering the model structure or optimization routine, evaluations on benchmarks like CIFAR-10 (+0.42%), CIFAR-100 (+0.99%), CIFAR10-DVS (+0.1%), and ImageNet-1K (+0.73%) demonstrated consistent accuracy gains, accompanied by a minor increase in computational overhead. Our code is available at `https://github.com/NeuroSyd/DA-SSDP`.

## 1 Introduction

Spiking neural networks (SNNs) are depicted as the third generation of artificial neural networks (Maass, 1997). Driven by the growing demand for energy-efficient computing and the rising interest in brain-inspired architectures, SNNs have gained traction in recent years with neuromorphic hardware and biologically

---

[*]Corresponding author.

grounded research communities because of their event-driven nature and potential for significantly lowering energy consumption (Amir et al., 2017; Davies et al., 2018; Friedmann et al., 2016; Merolla et al., 2014). By transmitting information through discrete spikes, SNNs enable sparse and asynchronous processing that aligns closely with the characteristics of brain-inspired hardware (Furber et al., 2014; Merolla et al., 2014). That said, their performance on complex and large-scale benchmarks still lags behind conventional artificial neural networks (ANNs) (Hu et al., 2021; Shi et al., 2024; Zhou et al., 2022). This gap has led researchers to explore whether architectural techniques from ANNs, such as attention mechanisms from transformers, can be adapted to improve performance in spiking models.

Efforts to combine spiking dynamics with transformer-style attention mechanisms have recently gained momentum (Yao et al., 2024; Zhou et al., 2022). Several research groups have started to rework components of the transformer architecture, particularly self-attention, to better fit the characteristics of spike-based computation. One early example is Spikformer (Zhou et al., 2022), which replaces operations like softmax with spike-compatible alternatives, resulting in competitive accuracy on image classification tasks. Building on this foundation, Yao et al. (2024) proposed a fully spike-driven transformer, while SpikingResformer (Shi et al., 2024) introduced residual convolutional blocks into a spiking attention framework, achieving 79.40% Top-1 accuracy on ImageNet using only four time steps, narrowing the gap between SNNs and standard ANNs.

Despite rapid architectural progress, current spiking transformers are trained almost exclusively with surrogate backpropagation. What is missing is a scalable, structure-aware local signal that can be injected during training without altering the architecture. Conventional regularizers (e.g., weight decay (Krogh & Hertz, 1991), label smoothing (Szegedy et al., 2016), dropout (Srivastava et al., 2014))operate on weights or real-valued activations and largely ignore the event structure that determines computation in SNNs. In contrast, timing-based plasticity rules accurately capture certain neural behaviors. However, when simply adapted to deep, multi-layer networks, they demand detailed temporal tracking and extra memory for states. Although Tian et al. (2025) provides a preliminary exploration of synchrony-based signals, its adjustments are not aligned with the supervised learning signal, which hinders a safe and scalable training-time implementation.

In contrast to most artificial systems, biological brains combine both supervised and unsupervised forms of learning (Storrs et al., 2021; Hennig et al., 2021). Synaptic changes are typically driven by local spike activity patterns (Feldman, 2012), and these changes can also be shaped by neuromodulatory signals. Dopamine, for instance, plays a central role in linking synaptic modification to reward feedback (Frémaux & Gerstner, 2016; Speranza et al., 2021). Inspired by this mechanism, we propose integrating biologically motivated, self-organizing plasticity into spiking transformers to enhance their learning capabilities and generalization. Previous research indicates that Hebbian plasticity (Hebb, 1949) can effectively support continual learning and unsupervised feature extraction in SNNs (Xiao et al., 2024). Yet, current spiking transformer implementations do not incorporate such biologically inspired learning mechanisms.

In this work, we present a novel training strategy for spiking transformers, integrating supervised gradient-based training with an unsupervised, reward-modulated plasticity mechanism. We name this method **D**opa**m**ine-modulated **S**pike-**S**ynchrony-**D**ependent **P**lasticity (**DA-SSDP**). The fundamental idea is to monitor spike-timing synchrony during training and dynamically adjust synaptic weights in response to task-specific loss. Essentially, DA-SSDP encourages synapses that contribute to synchronized spikes correlated with correct predictions, and penalizes those associated with poorly timed spikes, mimicking the modulatory role of dopamine observed in shaping neural coordination and learning. Through this additional plasticity mechanism, we aim to (i) capture subtle spike synchronous correlations potentially overlooked by gradient descent, (ii) regularize neural activity to reduce uncoordinated spiking, and (iii) explore how synchrony-based local plasticity learning mechanisms can be integrated with global learning signals to improve generalization without imposing additional computational cost.

This paper presents the following primary contributions:

(A) **Bringing synchrony-based signal into model learning:** DA-SSDP injects a batch-level, synchrony-based complementary learning signal during training. DA-SSDP departs from Spike-Timing-Dependent Plasticity (STDP) by screening for co-firing synchrony before any update; only

then is a bounded Hebbian adjustment applied, independent of pre/post order. A brief warm-up phase derives a dopamine-inspired scaling factor from synchrony and loss. After that, the gate stays fixed and simply adjusts the size of a Hebbian-style local update for each batch. When synchrony is task-irrelevant, the gate collapses to unity and the rule safely reverts to the two-factor baseline (no online loss), providing a scalable, structure-aware signal without architectural changes.

(B) **Scalable training-time synchrony update for deep SNN-transformers:** DA-SSDP requires only binary spike indicators and first-spike latencies, adding only $O(C_{\text{out}}C_{\text{in}})$, where $C_{\text{in}}$ refers to the number of pre-synaptic channels and $C_{\text{out}}$ refers to the number of post-synaptic channels of the module to which DA-SSDP is attached, with per-batch element-wise operations and no inference-time cost. We provide a drop-in implementation that tweaks just the final classification part and a simple late-stage feature projector, keeping the architecture and training schedule (epochs, batch size, optimizer) unchanged. This approach saves a lot of memory compared with storing detailed spike-timing records.

(C) **Quantitative validation and robustness analysis with safe degradation:** Across CIFAR-10/100, ImageNet-1K, and an event stream (CIFAR10-DVS), DA-SSDP shows consistent accuracy gains under the same training setup. On the test set, the median batch synchrony $S_b$ increases from $3 \times 10^{-4}$ to $1.0 \times 10^{-2}$ ($\sim 33 \times$).

## 2 Related Works

**Spiking transformers and training-time cost:** Recent SNN transformers (Zhou et al., 2022; 2023; Shi et al., 2024; Yao et al., 2024; Zhou et al., 2024a; Yao et al., 2024) primarily target inference-time efficiency by exploiting sparse spikes and a few time steps while maintaining competitive accuracy on image benchmarks. However, training remains almost entirely surrogate backprop (Zhou et al., 2024a; Gygax & Zenke, 2025; Hu et al., 2024), whose unrolling, state storage, and memory traffic can be comparable to, or even higher than ANN counterparts (Zhou et al., 2024b). Architectural progress, therefore, narrows the inference gap but does not by itself provide a scalable, structure-aware training signal that interacts with spike statistics.

**Local plasticity at scale:** Biologically motivated rules range from two-factor timing/trace updates to three-factor modulated variants (Mazurek et al., 2025). In deep, multi-layer neural networks, timing-based plasticity rules run into two ongoing problems. First, scalability is hindered by keeping detailed records of spike timings and assigning credit/blame between pairs of neurons over time, which consumes a large amount of memory and processing bandwidth. Secondly, the update directions are not coordinated with the supervised loss and may conflict with gradient signals. Consequently, most demonstrations remain on small-scale tasks or online/RL settings (Mazurek et al., 2025; Zhou et al., 2024b). Along two useful axes, local signal (pairwise timing *vs.* population synchrony) and global modulation (absent vs reward/prediction-error), prior work largely emphasizes pairwise timing, sometimes gated by a neuromodulator, while underusing population-level coordination. Yet neuroscience highlights population synchrony as a description of coordinated activity (Vinck et al., 2023; Majhi et al., 2024), whereas synchrony has rarely been used as a scalable, supervised training-time signal in deep SNNs.

We fill this gap by using batch-level spike synchrony as a low-memory substitute. It relies on binary spike indicators and first-spike latencies with $O(C_{\text{out}}C_{\text{in}})$ per batch. We then adjust its local influence using a dopamine-inspired gate, derived from synchrony and loss during a brief warm-up period. Once set, this modulation remains constant and merely scales a Hebbian-style local update. As a result, the rule enhances backpropagation without modifying the forward pass dynamics. When synchrony is uninformative, the gate converges to one, simplifying the update to a harmless two-factor baseline that's applied after the main updates in deeper layers. This provides the scalable training signal that prior spiking transformers have been missing. The formulation and implementation are detailed next.

## 3 Method

This section introduces the proposed DA-SSDP mechanism and its role in enhancing the training of spiking transformer models. We first describe the DA-SSDP rule itself, then explain how it is integrated into the

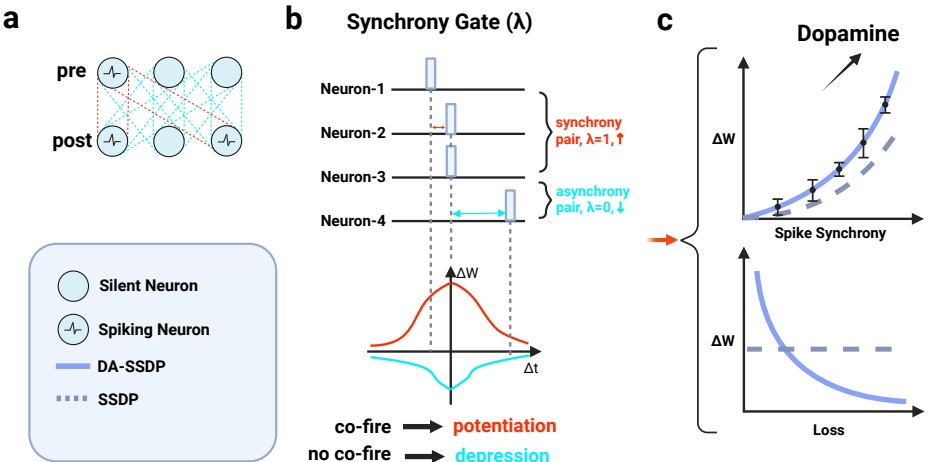

Figure 1: **Overview of DA-SSDP (a)** Pre–post layer and binary activity indicators from the current mini-batch. **(b) Synchrony gate** $\lambda$ **-** A pre/post pair that co-fires in the sample sets $\lambda = 1$ and is potentiated (red), no co-firing sets $\lambda = 0$ and yields depression (cyan). The first-spike lag $|\Delta t|$ is passed through a Gaussian window $g = \exp\left(-\Delta t^2/(2\sigma^2)\right)$, which scales the magnitude only and the rule is order-invariant (synchrony decides the sign, timing sets the strength). **(c) Dopamine modulation -** During a short warm-up, a single scalar slope is fit from the empirical synchrony–loss correlation and after warm-up, the slope is frozen. Thereafter, the gate depends only on batch synchrony and simply rescales the local SSDP update (solid blue: DA-SSDP; dashed: ungated SSDP).

SpikingResformer backbone and trained in a two-stage manner. Fig. 1 summarizes DA-SSDP. For each pre/post pair within a sample, a binary synchrony gate $\lambda \in \{0, 1\}$ indicates whether both neurons fired at least once, co-firing ($\lambda = 1$) leads to potentiation and no co-firing ($\lambda = 0$) leads to depression (Fig. 1(a,b)). The absolute difference in first-spike times, $|\Delta t|$, feeds into a Gaussian function $g = \exp(-\Delta t^2/(2\sigma^2))$ to adjust only the magnitude of the update. The sign comes from the synchrony itself, ensuring the rule does not depend on the order of spikes. Summing up $\lambda g_{b,i,j}$ across all connections produces a synchrony measure for the entire batch. During a short warm-up, we fit a single negative slope that captures the inverse relationship between synchrony and loss, and then freeze it. From then on, this factor serves as a scalar that rescales the local SSDP update without changing its direction (Fig. 1(c)).

## 3.1 Dopamine-Modulated SSDP Mechanism

We use a mini-batch of size $B$, where samples are indexed by $b \in \{1, \dots, B\}$. For a hooked module with $C_{\text{in}}$ input channels and $C_{\text{out}}$ output channels, we index pre-synaptic channels as $j \in \{1, \dots, C_{\text{in}}\}$ and postsynaptic channels as $i \in \{1, \dots, C_{\text{out}}\}$. Let $P$ be the binary pre-synaptic spike matrix of shape $P \in \{0, 1\}^{B \times C_{\text{in}}}$ and $Q$ the binary postsynaptic spike matrix of shape $Q \in \{0, 1\}^{B \times C_{\text{out}}}$ for a mini-batch of size $B$ (each row resembles a sample, and a column represents a channel). In the implementation, these indicators are obtained by thresholding layer activations during the forward pass and keeping only whether a channel fired at least once in the window, which yields a stable binary co-activation signal that is cheap to cache and process at scale. For every $(j, i)$ we also record the first-spike latency $\Delta t_{b,i,j} = \left| t_{b,i}^{\text{post}} - t_{b,j}^{\text{pre}} \right|$. Only the first spike time is stored, minimizing memory use while retaining the core temporal signal. Channels that remain silent throughout the window receive a $t = T$, where $T$ is the window length in time steps. The first spike has been shown to carry most of the critical information, whereas subsequent spikes contribute little additional content and may even introduce redundancy (Thorpe et al., 2001; van Rullen & Thorpe, 2001; Gollisch & Meister, 2008). The forward pass provides these timestamps directly as tensors $t^{\text{pre}} t^{\text{post}}$, enabling on-the-spot computation of $\Delta t$ without needing to rerun the sequence.

We now outline the fundamental update mechanism of DA-SSDP. For a mini-batch, the weight adjustment between pre-synaptic $j$ and and post-synaptic $i$ is given by

$$\Delta W_{i,j} = \text{clip}\left(\frac{1}{B}\sum_{b=1}^{B} G_b\, g_{b,i,j}\Big(A_+\lambda_{b,i,j} - A_-(1-\lambda_{b,i,j})\Big), -1, 1\right), \tag{1}$$

where $\lambda_{b,i,j} = Q_{b,i}P_{b,j} \in \{0,1\}$ signals co-spiking between the post- and pre-synaptic channels, $g_{b,i,j} = \exp\big[-\Delta t^2/(2\sigma^2)\big]$ is a Gaussian kernel with learnable bandwidth $\sigma > 0$, and $A_+, A_- > 0$ control the degrees of potentiation and depression, respectively. The batch-specific scalar gate $G_b$ is detailed further in the warm-up calibration section (Eq. 10).

This equation highlights three key elements of DA-SSDP. First, the **binary synchrony gate** $\lambda$ ensures that only coincident pre/post events contribute to plasticity, grounding the rule firmly in spike synchrony. Second, the **temporal kernel** $g(\Delta t)$ prioritizes updates for near synchronous events while suppressing those with larger delays, reflecting how spike timing influences synaptic strength. Third, the **Loss-aware modulation** $G_b$ injects a loss-dependent signal, enabling the local plasticity rule to remain compatible with global task objectives. Together, these elements produce weight increments that are clipped to a bounded range, preventing instability and ensuring stable training in deep architectures.

### 3.1.1 Instantaneous Potentiation and Depression

**Gaussian weighting:** For each pre–post pair, we apply a continuous Gaussian kernel

$$g_{b,i,j} = \exp\left[-\frac{\Delta t_{b,i,j}^2}{2\sigma^2}\right], \tag{2}$$

where $\sigma$ is a learnable bandwidth parameter that's shared across the module. Since $g$ decreases with $|\Delta t|$, updates are strongest for nearly synchronous spikes and taper off as the pre–post latency widens. Using a continuous kernel avoids fragile outcomes tied to rigid thresholds and maintains effective gradients during combined training with backpropagation.

**Synchrony gate:** Co-activation is represented by a binary mask

$$\lambda_{b,i,j} = Q_{b,i}P_{b,j} \in \{0,1\}. \tag{3}$$

Here $\lambda$ is computed via an outer product $QP^\top$ for each sample (i.e., for sample $b$, $Q_b \in \mathbb{R}^{C_\text{out}\times 1}$ and $P_b \in \mathbb{R}^{C_\text{in}\times 1}$; we transpose $P_b$ to $1\times C_\text{in}$ to form the outer product $Q_bP_b^\top$, yielding a $C_\text{out}\times C_\text{in}$ mask) at $O(C_\text{out}C_\text{in})$ complexity and no need for temporal scanning, because first-spike times are computed once per channel (not per pair) and $\Delta t$ is formed by broadcasting $t^\text{post}$ and $t^\text{pre}$, avoiding any $O(TC_\text{out}C_\text{in})$ per-pair search. Consequently, the combined factor $g_{b,i,j}\lambda_{b,i,j}$ functions as a gentle synchrony sensor, and it remains strictly positive only when the pair spikes together within the window, and its value adjusts according to temporal closeness through $g$.

**Weight update:** The per-sample plasticity change for a given sample $b$ and connection pair $(i,j)$ is defined as

$$\Delta w_{b,i,j} = g_{b,i,j}\Big((A_+ + A_-)\,\lambda_{b,i,j} - A_-\Big), \tag{4}$$

leading to potentiation when $\lambda = 1$ and depression when $\lambda = 0$. In cases where both units spike, $\lambda = 1 \Rightarrow \Delta w = +A_+ g$. Spikes that are closely timed ($\Delta t \approx 0$) produce the strongest potentiation, with bigger delays diminishing it. When only one unit spikes, $\lambda = 0 \Rightarrow \Delta w = -A_- g$. For both silent: $\lambda = 0$ and $t^\text{pre}=t^\text{post}=T \Rightarrow \Delta t = 0$, so $g = 1$ and $\Delta w = -A_-$. Coactive synapses are reinforced according to their temporal alignment, while inactive pairs are mildly suppressed, which limits accidental co-firing and improves generalization without maintaining eligibility traces for each synapse.

### 3.1.2 Batch-level Synchrony Score

We combine spike occurrences into a single scalar synchrony value

$$S_b = \frac{1}{C_{\text{out}}C_{\text{in}}} \sum_{i,j} Q_{b,i} P_{b,j} = \langle Q_b P_b^{\top} \rangle, \tag{5}$$

which ranges in $[0,1]$ and measures how synchronous the two populations are on sample $b$. In the implementation, this is calculated as the average of the outer product over the $(i,j)$ dimensions. The division by $C_{\text{out}}C_{\text{in}}$ ensures $S_b$ can be meaningfully compared between different layers or architectures with varying channel sizes. The definition is permutation-invariant with respect to channel ordering and depends only on binary co-firing, which makes it robust to activation scale and batch normalization statistics. Relying on co-firing rather than full $\Delta t$ details, the signal maintains low variance over time windows, while $\Delta t$ still influences the individual updates through the Gaussian kernel.

### 3.1.3 Warm-up and Gate Calibration

During the $E_{\text{warm}}$ epochs, DA-SSDP does not modify weights. Instead, it collects per-batch pairs $(S_b, \ell_b)$, where $S_b$ is the batch-wise synchrony metric and $\ell_b$ is the supervised loss for that batch. These pairs are aggregated across all warm-up batches to form two aligned sequences.

**Standardization:**  Let $\{S_b\}_{b=1}^N$ and $\{\ell_b\}_{b=1}^N$ represent the collected sequences from the warm-up phase (length $N$). We calculate their means and standard deviations as follows,

$$\mu_S = \frac{1}{N} \sum_{b=1}^N S_b, \qquad \sigma_S = \sqrt{\frac{1}{N} \sum_{b=1}^N (S_b - \mu_S)^2}, \tag{6}$$

$$\mu_\ell = \frac{1}{N} \sum_{b=1}^N \ell_b, \qquad \sigma_\ell = \sqrt{\frac{1}{N} \sum_{b=1}^N (\ell_b - \mu_\ell)^2}, \tag{7}$$

and derive the normalized values

$$\hat{S}_b = \frac{S_b - \mu_S}{\sigma_S}, \qquad \hat{\ell}_b = \frac{\ell_b - \mu_\ell}{\sigma_\ell}. \tag{8}$$

**Slope fitting:**  The dopamine slope is set by the negative empirical correlation between standardized synchrony and loss:

$$k = -\mathbb{E}_b[\hat{S}_b \, \hat{\ell}_b] = -\frac{1}{N} \sum_{b=1}^N \hat{S}_b \, \hat{\ell}_b, \tag{9}$$

where $\mathbb{E}_b[\cdot]$ signifies the empirical mean across the warm-up collection. In essence, a pattern of higher synchrony pairing with reduced loss produces $k > 0$, boosting weights for future high-$S_b$ batches. A feeble connection yields $k \approx 0$, rendering the gate essentially impartial.

**Gate at training time:**  After warm-up, $(\mu_S, \sigma_S, k)$ remain fixed. For each new batch, we compute the scalar gate.

$$G_b = \text{clip}\Big(1 + k \, \frac{S_b - \mu_S}{\sigma_S}, 0, 2\Big), \tag{10}$$

which is uniformly applied to all synaptic pairs $(i,j)$ within the batch, scaling the per-sample term in Eq. 1. The $\text{clip}(x,a,b) := \min\{\max\{x,a\}, b\}$ keeps the modulation bounded and prevents explosive adjustments.

**Robustness:**  If the warm-up data offers limited insight, for example, if the batch synchrony $S_b$ has a very small dynamic range (leading to small $\sigma_S$) or negligible correlation with loss (so $k \approx 0$), we fall back to a neutral gate via $k=0$ with standardized $S_b$. This securely deactivates the dopamine component without disrupting the overall training flow. For a practical demonstration, consider the CIFAR10-DVS example in Sec. 4.2, where $S_b$ exhibits minimal variation during warm-up, yielding a fitted slope $k \approx 0$ with $G_b \approx 1$.

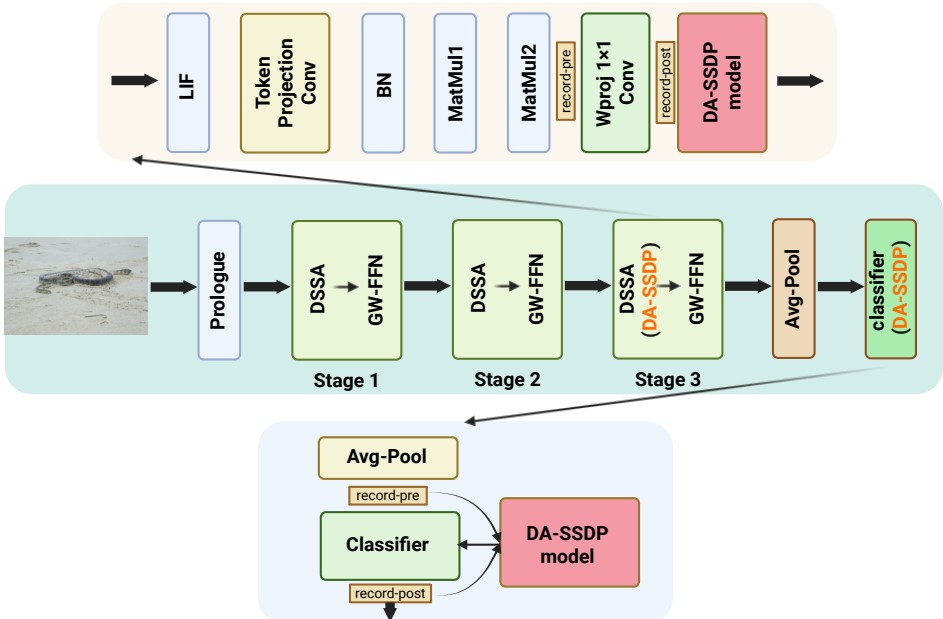

Figure 2: **DA-SSDP integration points in SpikingResformer -** The DA-SSDP module is inserted at two locations (1) the 1×1 projection convolution of the last DSSA block in Stage 3 (2) the linear classifier following global average pooling. During training, each hook records pre-/post-spike activity and applies the DA-SSDP update after the warm-up phase, operating alongside standard back-propagation.

### 3.1.4 Complexity and Stability

All DA-SSDP computations reduce to broadcast outer products, element-wise exponential for the Gaussian kernel, scalar standardization, and simple reductions over Boolean tensors. Overall, the effort grows as $\mathcal{O}(B\,C_{\text{out}}C_{\text{in}})$ for each mini-batch, primarily involving efficient vectorized tensor operations that align seamlessly with GPU acceleration. No synapse-specific eligibility traces persist between steps, with only first-spike timings and binary indicators stored from the forward pass; thus, the maximum memory burden consists of the batch-level accumulator $\Delta W$ alongside minor per-batch values $(S_b, \mu_S, \sigma_S, k)$. After calibration, the gate parameters $(k, \mu_S, \sigma_S)$ are fixed, whereas $A_+$, $A_-$, and $\sigma$ continue to be trainable. This separation ensures numerical reliability in the dopamine-modulated pathway and prevents fluctuations towards the end of training. Backpropagation continues as the core force behind convergence, while DA-SSDP contributes additional, task-relevant information by coupling local population synchrony $S_b$ to the global objective through the dopamine gate $G_b$. This coupling reinforces synchrony only when it predicts lower loss, guiding the network toward more stable firing regimes. Otherwise, the gate neutralizes, and the update reduces to the two-factor SSDP baseline.

### 3.2 Implementation Details

**Integration of DA-SSDP into SpikingResformer:** We integrated DA-SSDP into the SpikingResformer architecture, as illustrated in Fig. 2. The model follows a hierarchical backbone comprising (i) a front-end downsampling stem (Prologue) that maps the input image to an initial spike feature map, (ii) spiking self-attention layers (DSSA), (iii) grouped convolutional feed-forward blocks (GWFFN), and (iv) a spike-based classifier at the output stage. We insert two DA-SSDP modules as light-weight post-update hooks without changing the forward path (i) the linear classifier and (ii) the 1×1 projection in the last DSSA block. At this hook, we consider the DSSA input channels as presynaptic and the projection outputs as postsynaptic. In every training iteration, we begin with the usual surrogate-gradient update. Next, using the current

mini-batch, we generate binary activity markers and per-channel first-spike timings at the two hook locations, invoke DA-SSDP to calculate an additive weight adjustment $\Delta W$, and apply it directly to the relevant weights. During the warm-up phase, $e < E_{\text{warm}}$ where e denotes the current epoch and $E_{\text{warm}}$ denotes the maximum epoch of the warm-up phase for computing the DA gate, DA-SSDP only records statistics without making any weight modifications. For the classifier hook, we extract pre- or post-activity from the per-step readout activations, and for the DSSA hook, a channel counts as active in a time step if any of its spatial positions is active, with the first-event timing set to the earliest step where the indicator activates. The two DA-SSDP instances maintain separate parameters and warm-up data. This setup preserves attention maps and all forward-pass operations intact, with DA-SSDP simply providing a minor post-update adjustment driven by batch-level synchrony.

Table 1: Top-1 Classification accuracy, Values are reported as mean ($\pm$) standard deviation over five runs. Energy is the estimation of energy consumption, same as (Yao et al., 2024; Zhou et al., 2022; Shi et al., 2024) in Appendix A.2.

| Dataset | Method | Architecture | Param (M) | Energy (mJ) | TimeStep | Accuracy (%) |
|---|---|---|---|---|---|---|
| **CIFAR10** | Spikformer (Zhou et al., 2022) | Spikformer-4-384 | 9.32 | — | 4 | 95.51 |
| | Spikingformer (Zhou et al., 2023) | Spikingformer | — | — | 4 | 95.61 |
| | Spike-driven Transformer (Yao et al., 2024) | Spike-driven Transformer | — | — | 4 | 95.6 |
| | SpikingResformer (Shi et al., 2024) | SpikingResformer-CIFAR | 10.83 | — | 4 | 95.95 |
| | **SpikingResformer+SSDP(w/o DA)** | SpikingResformer-CIFAR | **10.83** | — | 4 | **96.15 ± 0.09** |
| | **SpikingResformer+DA-SSDP** | SpikingResformer-CIFAR | **10.83** | — | 4 | **96.22 ± 0.1** |
| **CIFAR10-DVS** | Spikformer | Spikformer-4-384 | 9.32 | — | 16 | 80.6 |
| | Spikingformer | Spikingformer | — | — | 16 | 81.3 |
| | Spike-driven Transformer | Spike-driven Transformer | — | — | 16 | 80.0 |
| | Transformer | Transformer-4-384 | 9.32 | — | 1 | 81.02 |
| | SpikingResformer | SpikingResformer-CIFAR | 17.31 | 2.403 | 10 | 84.4 |
| | **SpikingResformer+SSDP(w/o DA)** | SpikingResformer-CIFAR | **17.31** | **2.421** | 10 | **84.5 ± 0.1** |
| | **SpikingResformer+DA-SSDP** | SpikingResformer-CIFAR | **17.31** | **2.456** | 10 | **84.5 ± 0.1** |
| **CIFAR100** | Spikformer | Spikformer-4-384 | 9.32 | — | 4 | 77.86 |
| | Spikingformer | Spikingformer | — | — | 4 | 79.09 |
| | Spike-driven Transformer | Spike-driven Transformer | — | — | 4 | 78.4 |
| | Transformer | Transformer-4-384 | 9.32 | — | 1 | 81.02 |
| | SpikingResformer | SpikingResformer-CIFAR | 10.83 | 0.493 | 4 | 78.73 |
| | **SpikingResformer+SSDP(w/o DA)** | SpikingResformer-CIFAR | **10.83** | **0.504** | 4 | **79.26 ± 0.22** |
| | **SpikingResformer+DA-SSDP** | SpikingResformer-CIFAR | **10.83** | **0.494** | 4 | **79.48 ± 0.24** |
| **ImageNet** | Spiking ResNet (Hu et al., 2021) | ResNet-50 | 25.56 | 70.934 | 350 | 72.75 |
| | Transformer (Dosovitskiy et al., 2021) | Transformer-8-512 | 29.68 | 38.340 | 1 | 80.80 |
| | Spikformer | Spikformer-8-384 | 16.81 | 7.734 | 4 | 70.24 |
| | Spikformer | Spikformer-8-768 | 66.34 | 21.477 | 4 | 74.81 |
| | Spikingformer | Spikingformer-8-512 | 29.68 | 7.46 | 4 | 74.79 |
| | Spikingformer | Spikingformer-8-768 | 66.34 | 13.68 | 4 | 75.85 |
| | Spike-driven Transformer | Spike-driven Transformer-8-768 | 66.34 | 6.09 | 4 | 76.32 |
| | SpikingResformer | SpikingResformer-L | 60.38 | 8.76 | 4 | 78.77 |
| | **SpikingResformer+SSDP(w/o DA)** | SpikingResformer-L | **60.38** | **8.89** | 4 | **79.12 ± 0.23** |
| | **SpikingResformer+DA-SSDP** | SpikingResformer-L | **60.38** | **8.76** | 4 | **79.29 ± 0.21** |

## 4 Experiments

We design our experiments to systematically evaluate the effectiveness of DA-SSDP in improving performance, refining temporal feature encoding, and energy cost in spiking transformer models. All experiments are conducted using publicly available vision datasets. Each experiment is repeated five times to estimate the variance. To maintain fairness, we preserve identical model structures and training regimens across variants, except where otherwise specified.

### 4.1 Experimental Setup

Our primary baseline is SpikingResformer (Shi et al., 2024). We keep the architecture and training schedule identical to this baseline and only attach DA-SSDP during training, so the comparison isolates the effect of our method. For a broader context, we also list representative results from earlier SNN-transformer variants (Zhou et al., 2023; 2022; Yao et al., 2024; Shi et al., 2024) as published. DA-SSDP uses a warm-up of $E_{\text{warm}}$=100 (for CIFAR10-DVS, $E_{\text{warm}}$=80) epochs to fit the gate, and the fitted gate is then kept fixed for the remaining epochs. Kernel parameters are $A_+$=1.5×$10^{-3}$, $A_-$=1.0×$10^{-4}$, and a learnable $\sigma$. Backbones follow Spikingresformer with $T$=4 steps on CIFAR-10/100 (Krizhevsky et al., 2009) and ImageNet-1k (Deng

et al., 2009), and $T$=10 on CIFAR10-DVS (Li et al., 2017). More training details are in Appendix A.1. Each configuration is repeated five times with different seeds.

## 4.2 Results

Table 1 summarizes the results. With identical model structures and comparable training configurations, DA-SSDP yields consistent performance boosts: +0.42% on CIFAR-10, +0.99% on CIFAR-100, and +0.73% on ImageNet-1k. Model size and inference cost are unchanged; the rule adds only train-time computation at the hooked layers. We further isolate the regularization effect with a 200-epoch (100-epoch warm-up + 100-epoch gated phase); see Appendix A.3 for gap and validation-advantage analyses and Appendix A.9 for further experiments.

**On CIFAR10-DVS:**  CIFAR10-DVS is created by showing static CIFAR-10 images to a DVS sensor while moving the screen or camera, which produces event streams dominated by edge-related contrast changes. After the first convolution and pooling layers, the neural activity becomes quite sparse in time, and many pre- and post-channel pairs never fire together within the $T$ steps, so the synchrony gate $\lambda$ is often zero. Even when both sides do fire, their first spikes are usually far apart, making $|\Delta t|$ large and the Gaussian weight $g(\Delta t)$ close to zero. Channels that stay silent contribute small negative updates through the $-A_- g$ term, which pushes weights down even further. At the batch level, the synchrony score $S_b$ varies only slightly across batches, so its correlation with the loss is weak. This makes the fitted slope $k$ close to zero, and the dopamine gate essentially neutral ($G_b \approx 1$). In practice, DA-SSDP then behaves almost like its basic two-factor version, with only small net updates. This explains why the improvement on CIFAR10-DVS is modest compared to frame-based datasets, and shows that the rule is most effective when the data provide strong, task-relevant spike synchrony.

## 4.3 Ablation Study

To quantify the contribution of the proposed rule, we compare

- **Baseline:** the spiking transformer trained only with surrogate backprop.

- **SSDP (two–factor):** local synchrony–based updates without dopamine gating.

- **DA-SSDP (three–factor):** the full method with a dopamine-modulated gate.

### 4.3.1 DA-SSDP Integration in the Model

We further examine the rule's optimal placement by applying it to (i) the linear classifier only, (ii) the DSSA block only, or (iii) both components at the same time. No further hyperparameter adjustments are made for each configuration.

Under identical hyperparameters, the baseline reaches 78.73% Top-1. Table 2 reports the effect of where the local rule is attached. Applying the update in the Prologue convolution destabilized training in our setting (diverged under the same schedule). By contrast, attaching the rule to the last DSSA block yields the largest gain (+0.58% for two–factor SSDP; +0.87% with the dopamine gate), while adding it only to the linear classifier gives a smaller but positive gain (+0.23%). Using both hooks gives the best result (79.72%, +0.99%). No extra tuning was performed per placement, and the warm-up–fitted gate is kept fixed thereafter.

These observations are consistent with the mechanism of DA-SSDP. The rule acts as a post-update, synchrony-based local adjustment whose benefit depends on the presence of task-aligned co-activation. Such co-activation is more reliably expressed in mid-to-late representations than in very early feature extractors, where activity is sparser and more input-driven, which explains why the last DSSA benefits most while the Prologue is unstable under the same hyperparameters. Across placements, the dopamine gate further improves over the two-factor variant, indicating that scaling the local update by the synchrony–loss relationship learned during warm-up is helpful for learning. Overall, DA-SSDP enhances model performance by injecting local synchrony signals into deeper layers and scaling them through the correlation with the global backpropagation loss.

Table 2: Effects of integrating SSDP into different parts of the model on the CIFAR-100 dataset. Adding SSDP to the DSSA module yields the largest accuracy gain (+0.58%), whereas integrating it into the classifier improves performance by +0.23%. When placed in the Prologue, however, the model fails to converge.

| Position | Accuracy (%) | Improvement (%) |
|---|---|---|
| Baseline (No SSDP) | 78.73 | — |
| Prologue+SSDP (w/o DA) | — | Divergent |
| Prologue+DA-SSDP | — | Divergent |
| DSSA+SSDP (w/o DA) | 79.31 | +0.58 |
| DSSA+DA-SSDP | 79.60 | +0.87 |
| Classifier+SSDP (w/o DA) | 78.96 | +0.23 |
| Classifier+DA-SSDP | 79.12 | +0.39 |
| DSSA+Classifier+SSDP (w/o DA) | 79.48 | +0.75 |
| DSSA+Classifier+DA-SSDP | 79.72 | +0.99 |

Table 3: Performance under different $A_+$ and $A_-$ settings. The highest improvement is achieved with $A_+ = 0.00015$ and $A_- = 0.00005$.

| DA | $A_+$ | $A_-$ | Accuracy (%) |
|---|---|---|---|
| ✗ | 0.0015 | 0.0005 | 78.86 |
| ✗ | 0.00015 | 0.00005 | 79.48 |
| ✗ | 0.00010 | 0.00010 | 79.03 |
| ✗ | 0.001 | 0.001 | Divergent |
| ✓ | 0.0015 | 0.0005 | **79.72** |
| ✓ | 0.00015 | 0.00005 | **79.40** |
| ✓ | 0.00015 | 0.0001 | **79.22** |
| ✓ | 0.001 | 0.001 | **79.21** |

### 4.3.2 Impact of SSDP parameters on Convergence and Performance

To better understand how DA-SSDP handles parameter variation, we compared it against both the baseline and a variant using SSDP alone. We vary the potentiation/depression amplitudes $(A_+, A_-)$ and compare the two–factor SSDP to the three–factor DA-SSDP with a fixed gate (learned during warm-up and then frozen). No additional hyperparameter retuning is performed per setting.

**Hyperparameter Sensitivity to** $(A_+, A_-)$**:** Table 3 highlights three key patterns (i) under high amplitudes $(A_+{=}A_-{=}10^{-3})$, the two-factor SSDP fails to converge, but DA-SSDP achieves 79.21% accuracy, demonstrating better stability with extreme hyperparameter; (ii) at moderate amplitudes, DA-SSDP matches or slightly exceeds SSDP (e.g., 79.48% *vs.* 79.40% at $1.5{\times}10^{-4}/5{\times}10^{-5}$) (iii) the best accuracy is obtained with a larger potentiation setting $(A_+{=}1.5{\times}10^{-3}, A_-{=}5{\times}10^{-4})$, where DA-SSDP reaches **79.72%** versus **78.86%** for SSDP.

**Interpretation:** These findings align with the dopamine gate's function in our implementation. After warm-up, it delivers a stable, batch-specific rescaling $G_b{=}1 + k\,\hat{S}_b$, for the local adjustment, which adjust the per-batch update magnitude and reduces sensitivity to overly large $(A_+, A_-)$ without changing the update direction or the forward computation. When $k{\approx}0$ (weak synchrony-loss correlation), the method reduces to the two-factor baseline. Overall, DA-SSDP enables safe use of larger potentiation/depression amplitudes and achieves equivalent or better accuracy within an identical training configuration.

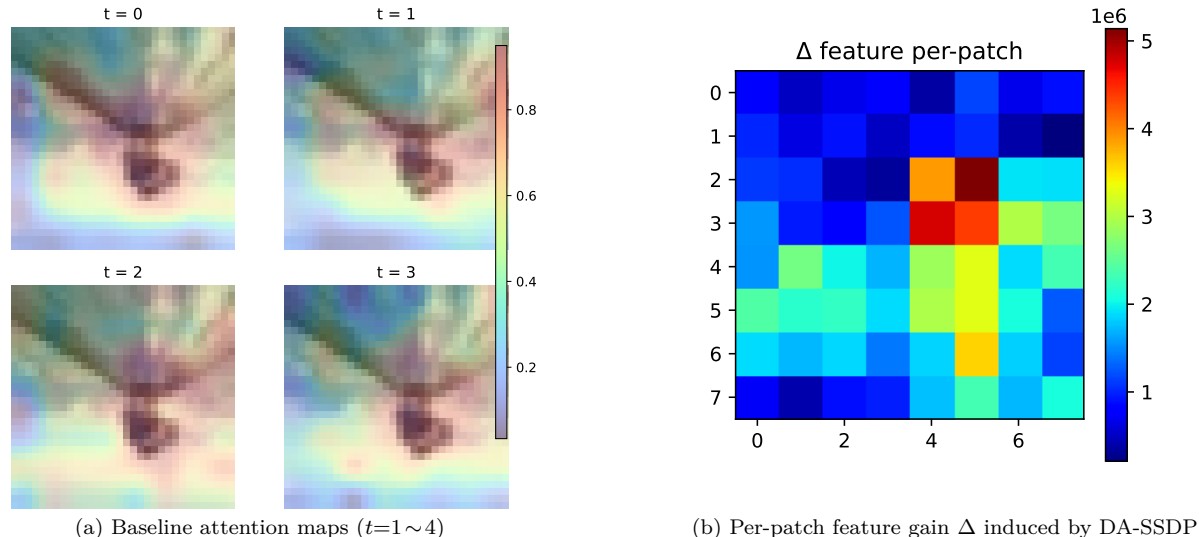

(a) Baseline attention maps ($t{=}1{\sim}4$)   (b) Per-patch feature gain $\Delta$ induced by DA-SSDP

Figure 3: **Spatial effect of DA-SSDP in the last DSSA stage** (a) Baseline attention maps (averaged over heads; $t{=}1{\sim}4$) on the same test image. (b) Per-patch feature-gain $\Delta(u,v)$, computed as the absolute change of the post-projection activation between the DA-SSDP and baseline checkpoints and averaged over time and channels (brighter = larger change). Hotspots in $\Delta$ align with high-attention areas from (a), suggesting a re-weighting of already attended tokens rather than a wholesale redirection.

### 4.4 Visualization and Activity Analysis

To explore the effect of DA-SSDP on temporal feature organization, we used several types of analysis. A qualitative t-SNE view of the test embeddings, with setup and plots, is provided in Appendix A.4.

**Spatial feature gain analysis:** We examine how DA-SSDP modifies spatial evidence within the final DSSA block using a typical test image. Fig. 3(a) shows the baseline attention maps (averaged across heads and time steps). Fig. 3(b) depicts a per-patch feature gain map, calculated as the absolute difference in post-projection activations between the final DA-SSDP model and the baseline version, as detailed in Eq. 11.

$$\Delta(u,v) = \frac{1}{TC} \sum_{t,c} \left| \left( W_{\text{proj}}^{\text{DA}} A_t^{\text{DA}} V_t^{\text{DA}} \right)_c (u,v) - \left( W_{\text{proj}}^{\text{Base}} A_t^{\text{Base}} V_t^{\text{Base}} \right)_c (u,v) \right|. \tag{11}$$

Hotspots in $\Delta(u,v)$ correspond to high attention zones in Fig. 3(a), aligning with our design. The local update connects to the attention layer, thereby preferentially recalibrating tokens with already prominent signals. In our implementation, DA-SSDP is attached only to the last-stage 1×1 projection $W_{\text{proj}}$ (and to the classifier); the attention operators remain unchanged. The update is computed at the channel level after aggregating spikes across spatial locations, so the gate $G_b$ and the synchrony signal do not encode where in the image a spike occurred. Consequently, the spatial map $\Delta(u,v)$ is larger in regions where the pre-projection features $A_t V_t$ already have higher magnitude (i.e., higher attention mass): the learned change in $W_{\text{proj}}$ reweights the existing channel mixture, and this effect is amplified where features are strong. This explains the co-location of hotspots with object regions in CIFAR-100 and supports our claim that DA-SSDP re-weights already attended tokens rather than redirecting attention.

**Spike Activity and Rate Analysis:** Fig. 4 demonstrates how DA-SSDP reorganizes the network's spiking behaviors. In the baseline network, spike frequencies vary broadly: many neurons fire two or three times within the four-step window, and a small subset fire at all four steps, leading to dense spike patterns. After introducing DA-SSDP, neurons crucial for decisions become more prone to simultaneous activation, and as a result, these channels are reinforced and maintain elevated spiking rates, while most other neurons emit at

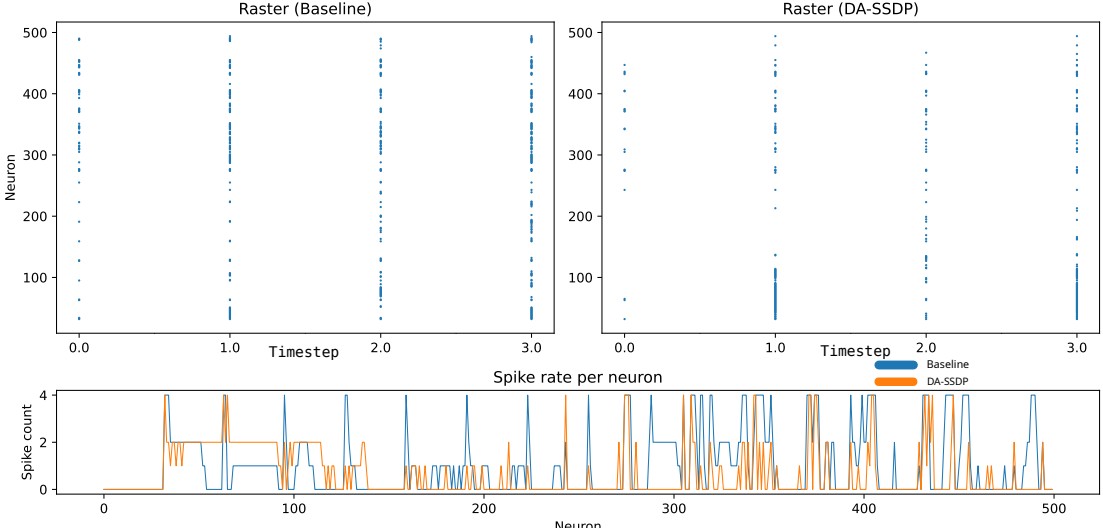

Figure 4: **Temporal spike patterns shaped by DA-SSDP -** Raster plots (top) and per–neuron spike counts (bottom) for the first 500 LIF/PLIF channels over $T = 4$ time steps, evaluated on a representative CIFAR-100 sample. Baseline activity is dense and widely spread, with many neurons firing on several of the four time steps and a small subset reaching the maximum count of four spikes. After DA-SSDP training, a compact synchronous burst emerges around indices 30–120, whereas the majority of channels emit at most a single spike or remain silent, yielding a markedly sparser profile.

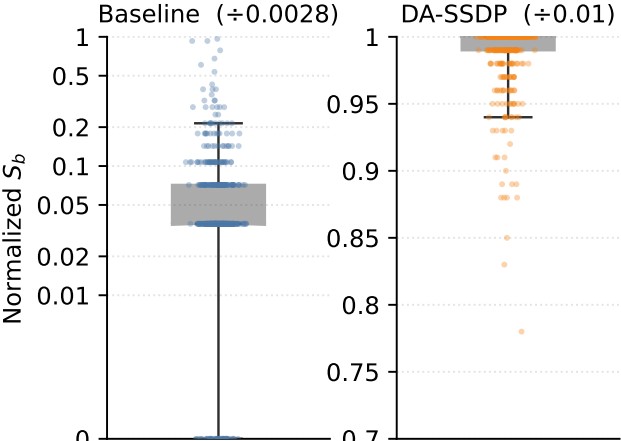

Figure 5: **Batch-level synchrony boxplot** Each dot is one mini-batch. For batch $b$, we mark a channel active if it fires at least once within the window $T$, and define the batch synchrony score $S_b$ as the fraction of channel pairs that spike in the same time bin. Relative to the vanilla baseline, DA-SSDP shifts the distribution upward and tightens its spread (median $3{\times}10^{-4} \to 1{\times}10^{-2}$, $\approx 33\times$). This pattern is consistent with the loss-aware gate $G_b$, which promotes task-aligned spike coincidences and suppresses asynchronous activity, leading to more coordinated and stable population activity.

most one spike or stay inactive. Taken together with the results in Table 1, DA-SSDP strengthens neurons that convey useful information while suppressing those offering minimal value or just adding noise.

These results confirm that the proposed plasticity rule not only reshapes temporal dynamics but does so in a manner that is aligned with the model's performance.

**Loss-aligned modulation via synchrony covariance:**  Using the gate $G_b$ in Eq. 10 with the slope $k$ from Eq. 9, DA-SSDP is expected to up-weight batches whose synchrony $S_b$ predicts lower loss and to stay neutral otherwise. Fig. 5 provides the mechanism-level signature of this behavior. Relative to the vanilla model, the entire distribution of $S_b$ on the test set shifts upward and contracts, with the median increasing from $3 \times 10^{-4}$ to $1 \times 10^{-2}$ ($\approx 33 \times$). This pattern indicates (i) more frequent task-aligned co-activation and (ii) reduced batch-to-batch variability in synchrony. Across random seeds, we observe the same qualitative shift. Taken together, the plot validates that the learned gate injects population-level information into the update stream and steers the network toward temporally stable firing regimes.

## 5  Discussion and Conclusion

**Loss-modulated synchrony as a scalable training-time signal.**  DA-SSDP incorporates a group-level signal into the training updates. Batch synchrony $S_b$ acts as a rough gauge of task-relevant simultaneous firing and impacts training exclusively via the gate $G_b$ and is inherently tuned to the supervised goal. As such, backpropagation continues as the core optimizer, whereas DA-SSDP adds a supportive inclination towards reliable, low-noise spiking dynamics in channels prone to join activation on well-predicted examples. This inclination disappears at inference time and preserves the forward-pass operations intact.

**Effect size tracks task-aligned co-activation in deeper layers.**  The rule is most effective in mid to late representations where task-aligned co-activation is reliably expressed. Applying the update to very early layers can be brittle under the same hyperparameters, consistent with sparser, input-driven activity in those stages. The dopamine-like gate fitted during warm-up rescales the local update per batch, which improves stability and enables the safe use of larger potentiation and depression amplitudes.

**Boundary conditions and stability under weak synchrony:**  The gains rely on the availability of synchrony linked to class-specific patterns. When the synchrony-loss correlation is weak, as observed on CIFAR10-DVS, the fitted slope collapses toward zero, the gate becomes neutral, and DA-SSDP reduces to its two-factor baseline with minimal effect. These properties clarify where the mechanism proves useful and ensure a safe degradation when synchrony offers little insight.

**Deployment in neuromorphic hardware or custom ASICs.**  DA-SSDP serves as a training-time regularizer and is ideally suited for offline use on GPUs to generate fixed weights ready for deployment. Although a straightforward silicon implementation would demand floating-point operations, latency storage, and custom accumulators, these components align well with neuromorphic hardware concepts. Local aspects, like detecting spike coincidences, function on a per-synapse or per-neuron-pair basis, relying on straightforward binary spike markers and time offsets—these can be processed efficiently on neuromorphic processors, capitalizing on their built-in capabilities for event-triggered synaptic modifications. Additionally, the global modulators resemble biological neuromodulatory transmissions, like dopamine releases, which can theoretically be realized and optimized on ASICs through basic scalar multiplications. By concentrating spikes into fewer, more synchronized channels and reducing incidental activity, the learned weights can indirectly improve efficiency on event-driven hardware. Detailed assumptions and quantitative estimates are provided in Appendix A.5.

**Limitations and future directions.**  While DA-SSDP shows promising improvements in accuracy, temporal coding, and estimated energy efficiency, a number of limitations remain. We outline some limitations of our proposed DA-SSDP, which we currently have under investigation and improvement.

(A) While our experiments demonstrate performance gains on standard vision benchmarks (reported as mean and standard deviation over five runs), a broader validation with larger sample sizes is needed to strictly establish statistical significance. In particular, evaluations on neuromorphic hardware platforms like Loihi or SpiNNaker could provide stronger evidence of generalization and practical utility.

(B) On datasets such as DVS, spike counts are extremely sparse and often distributed across wide temporal spans. Reliable performance typically requires a larger $T$ to capture sufficient information. In such regimes, the synchrony signal of DA-SSDP is suppressed by the Gaussian kernel, leading to near-zero incremental benefit over weights.

(C) We acknowledge that while DA-SSDP is motivated by biological plausibility, the introduction of a direct loss-derived global signal diminishes strict biological alignment. An interesting direction for future work is to explore alternative modulatory signals inspired by biological dopaminergic pathways, which may better balance biological compatibility with task-driven optimization (Kurniawan et al., 2011).

Future work includes extending synchrony-aware plasticity to self-supervised or pretraining regimes, exploring additional hook placements in attention pathways, and designing curriculum schedules that shape synchrony over time. A second direction is to couple DA-SSDP with online neuromodulatory signals in continual or reinforcement learning scenarios. Finally, we plan to quantify hardware effects on real accelerators by measuring event traffic, latency, and energy under fixed-weight deployment.

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

# A Appendix

## A.1 Training Configurations

**Setup and reproducibility:** All experiments were run from a unified training script. The setup involved a workstation with dual NVIDIA RTX 3090 GPUs, employing PyTorch 1.12.1 with CUDA 11.3 and NumPy 1.24.4.

**Model variants and spike simulation:** We use the `Spikingresformer` architecture with time steps $T$ set per dataset. Variants are chosen to match the input resolution: `spikingresformer_cifar` for CIFAR-100 and CIFAR10-DVS, and `spikingresformer_l` for ImageNet-1K.

**Datasets and preprocessing:** CIFAR-10/100 and ImageNet-1K use bicubic interpolation and per-dataset normalization. CIFAR-10 uses mean $(0.4914, 0.4822, 0.4465)$ and std $(0.2023, 0.1994, 0.2010)$, CIFAR-100 follows the common statistics from prior work and ImageNet-1K uses mean $(0.485, 0.456, 0.406)$ and std $(0.229, 0.224, 0.225)$. Event-based CIFAR10-DVS is loaded from SpikingJelly in frame representation with $T$ frames per sample and split 90%/10% for train/test. Frames are resized to the configured input size and optionally passed through a DVS-specific augmentation pipeline.

**Data augmentation and regularization:** For static image classification, we incorporate RandAugment and, if activated, Random Erasing along with mixup. CIFAR-100 employs the rand-m7-n1-mstd0.5-inc1 configuration, while ImageNet-1K utilizes rand-m9-n3-mstd0.5-inc1. Random Erasing operates at a 0.25 probability in constant mode whenever cutout is enabled. Mixup is handled with FastCollateMixup. Label smoothing of 0.1 is applied in the absence of soft labels. For CIFAR10-DVS, a lightweight DVS augmentation is used upon request.

**Optimization and schedules:** All runs use AdamW with the dataset-specific learning rate and weight decay reported below. A cosine learning-rate schedule is used for the entire training with a 3-epoch warm-up from 1e−5 and a minimum learning rate of 1e−5. We add a 10-epoch cooldown at the end of training. Unless otherwise stated, no gradient clipping or accumulation is used.

**Losses:** When mixup is enabled, we use SoftTargetCrossEntropy. Otherwise, we use cross-entropy with label smoothing. Losses are wrapped in the script's CriterionWrapper. Temporal Efficient Training (TET) is wired but disabled in all reported runs.

**Evaluation and compute reporting:** Top-1 accuracy is computed on the validation set after every epoch. MACs are measured with `thop` using custom counters for convolution, linear, and spike-aware matrix multiply. SOPs are monitored during inference with `SOPMonitor`. For step-mode `s` models, reported SOPs are multiplied by $T$ for comparability.

**Per-dataset hyperparameters:**

- **CIFAR-100:** Model: spikingresformer_cifar; input: $3{\times}32{\times}32$; epochs: 600; batch size: 200; $T$: 4; optimizer: AdamW with lr = 5e−4 and weight decay 0.01; augmentation: RandAugment rand-m7-n1-mstd0.5-inc1; mixup: on; cutout: off; label smoothing: 0.1; AMP: on; SyncBN: off.

- **CIFAR10-DVS:** Model: spikingresformer_s; input: 3×128×128; epochs: 100; batch size: 64; $T$: 10 frames; optimizer: AdamW with lr = 4e−5 and weight decay 0.01; mixup: on; cutout: off; label smoothing: 0.1; AMP: on; SyncBN: off.

- **ImageNet-1K:** Model: spikingresformer_l; input: 3×224×224; epochs: 320; batch size: 16; $T$: 4; optimizer: AdamW with lr = 5e−4 and weight decay 0.01; augmentation: RandAugment rand-m9-n3-mstd0.5-inc1 with mixup $\alpha$=0.2 and cutmix $\alpha$=1.0; label smoothing: 0.1; AMP: on; SyncBN: on.

**Implementation notes:** For CIFAR experiments, the data loader applies bicubic interpolation, random horizontal flipping, and skips extra cropping aside from RandAugment. ImageNet employs random resized cropping with a scale range of $[0.08, 1.0]$ and aspect ratios between $[3/4, 4/3]$. For DVS setups, the training sampler accounts for distributed processing, while the test sampler proceeds sequentially. Gradients are cleared at each iteration, and spiking states are reinitialized following the synchrony adjustment. During training, we store checkpoints for both the highest Top-1 performing model and the most recent one.

### A.2  Computation of MACs, SOPs, and Estimated Energy

We follow the standard accounting used in prior SNN works (Yao et al., 2024; Zhou et al., 2022; Shi et al., 2024). All quantities are reported per image, per inference and $T$ denotes the number of simulation steps.

**Synaptic operations.** For a block or layer $l$ in SpikingResformer, the synaptic operation count is related to its static arithmetic cost and the activity level of its inputs:

$$\text{SOPs}(l) \approx \bar{f}_r(l) \times T \times \text{MACs}(l), \tag{12}$$

where $\bar{f}_r(l)$ is the average input firing rate of block $l$, $T$ is the simulation length, and $\text{MACs}(l)$ is the multiply–accumulate count of $l$ obtained by structural profiling (`thop` with custom counters). In our implementation, $\text{SOPs}(l)$ are measured dynamically by a forward-hook monitor, which is consistent with the scaling in Eq. 12.

**Model-level estimated energy.** Assuming per-operation energy coefficients on a $45\,\text{nm}$ process, $E_{\text{MAC}} = 4.6\,\text{pJ/MAC}$ and $E_{\text{AC}} = 0.9\,\text{pJ/SOP}$, the energy estimate of SpikingResformer is

$$
\begin{aligned}
E_{SpikingResformer} = {} & E_{\text{MAC}} \cdot \text{MACs}\left(\text{FL}^1_{\text{SNN-Conv}}\right) \\
& + E_{\text{AC}} \cdot \left( \sum_{n=2}^{N} \text{SOPs}^n_{\text{SNN-Conv}} + \sum_{m=1}^{M} \text{SOPs}^m_{\text{SNN-FC}} + \sum_{\ell=1}^{L} \text{SOPs}^\ell_{\text{DSSA}} \right),
\end{aligned} \tag{13}
$$

where $\text{FL}^1_{\text{SNN-Conv}}$ denotes the first convolution that converts the static RGB image into spikes; the subsequent spiking convolution blocks, spiking fully-connected blocks, and DSSA blocks contribute activity-dependent SOPs.

Table 4: SOPs and estimated (Est) energy per image (Img) on *SpikingResformer–CIFAR*. Estimated energy follows the mapping described in Appendix A.2.

| Method | SOPs (G) | Est. energy per image ($\mu$J/Img) | $\Delta(\%)$($\mu$J/Img) *vs.* Base |
|---|---|---|---|
| Baseline | 0.54846 | 493.62 | − |
| DA-SSDP | 0.56425 | 494.82 | +0.24 |

**Block-wise form.** For an arbitrary block $b$,

$$\text{Estimated energy}_{\text{ANN}}(b)/\text{s} = 4.6\,\text{pJ} \times \text{MACs}(b), \qquad \text{Estimated energy}_{\text{SNN}}(b)/\text{s} = 0.9\,\text{pJ} \times \text{SOPs}(b), \tag{14}$$

where $\text{MACs}(b)$ or $\text{SOPs}(b)$ are expressed in billions (G), which results in the mJ/s estimation of energy.

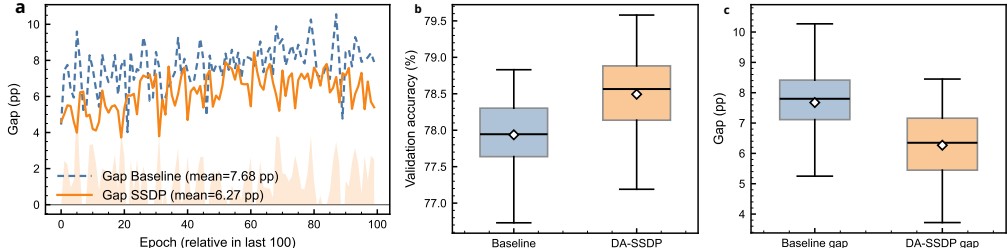

Figure 6: **DA-SSDP reduces the generalization gap and improves validation accuracy.** We trained the same CIFAR-100 model for 200 epochs with identical hyperparameters and seeds. The first 100 epochs served as a warm-up during which the DA-SSDP gate learned a slope linking batch-level synchrony to loss reduction; no DA-SSDP weight updates were applied in this phase. In the last 100 epochs, DA-SSDP was enabled alongside backpropagation. **(a)** SSDP yields a smaller generalization gap (training minus validation accuracy) across most epochs, with the mean gap reduced from 7.68 to 6.27 percentage points. **(b)** Validation accuracy is higher under DA-SSDP (mean 78.49% vs. 77.94%). **(c)** The gap distribution shifts lower under DA-SSDP, consistent with milder training fit and better generalization.

## A.3 Regularization Effects of DA-SSDP

We trained the same CIFAR-100 model for 200 epochs with the same hyperparameters and seeds. The first 100 epochs served as a warm-up period, during which the DA-SSDP gate learned a slope that links batch-level synchrony to loss reduction. However, no update was applied to the weights, and so accuracy did not change. In the remaining 100 epochs, we turned on DA-SSDP as a training-time update alongside backpropagation as showen in Fig.6.

Across the last 100 epochs, training accuracy with DA-SSDP was a bit lower than the baseline, while validation accuracy was higher through the mid-to-late stage and at the end. DA-SSDP won with an average validation gain of about 0.56%. The generalization gap, defined as training minus validation accuracy per epoch, dropped from about 7.68 to about 6.27 percentage points over the same window. We observed the same trend across seeds.

These findings confirm that DA-SSDP serves as a regularizer by modifying weights based on synchrony signals, encouraging learning of feature groups that consistently fire together while avoiding easily overfitted patterns that lack generalization. This leads to less aggressive training data fitting, a smaller gap between training and validation curves, and a small but steady increase in validation and test accuracy.

## A.4 Additional Visualization: t-SNE of Test Embeddings

Fig. 7 compares the hidden state embeddings across the two models. In the baseline model, each class clusters into a compact, circular hub, resulting in wide, vacant spaces between hubs. It results in a configuration that commonly signals overfitting to the "simpler" examples near class centroids. On the other hand, in DA-SSDP, after applying the adaptive gate in Eq. 9 and Eq. 10, the original compact clusters extend into wider, sometimes stretched shapes. Samples belonging to the same class remain grouped, but sparse connections occasionally form between similar categories, creating smoother, rather than sharp class boundaries. Both panels use the same t-SNE configuration (identical hyperparameters and random seed) applied to the same test features, and thus the differences reflect the learned representations rather than projection randomness.

The reason is that the gate only adjusts the strength of the original SSDP update, and it enhances potentiation for samples showing higher synchrony and lower loss ($S_b > \mu_S$ and $k > 0$), while for samples with below-average synchrony, it reduces both potentiation and depression. As a result, the network reinforces typical firing patterns without forcing all data points into overly tight clusters. Samples near decision boundaries or with unusual features receive gentler updates, creating a more flexible decision boundary. Although the resulting geometry appears less organized in two dimensions, it corresponds to the improved Top-1 accuracy

achieved by DA-SSDP (79.66% *vs.* 78.73% baseline). In essence, this seeming drop in visual tightness embodies a synchrony-driven regularization that enhances the model's ability to generalize.

**a**

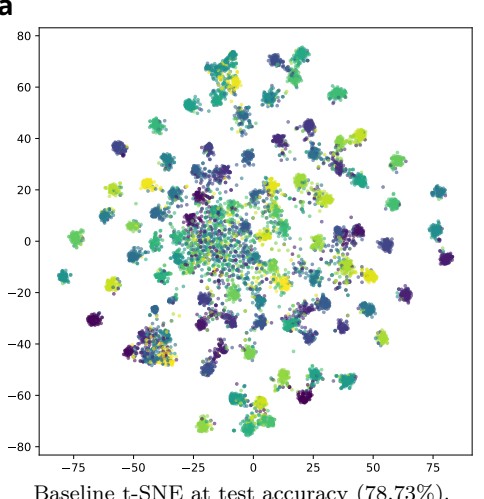

Baseline t-SNE at test accuracy (78.73%).

**b**

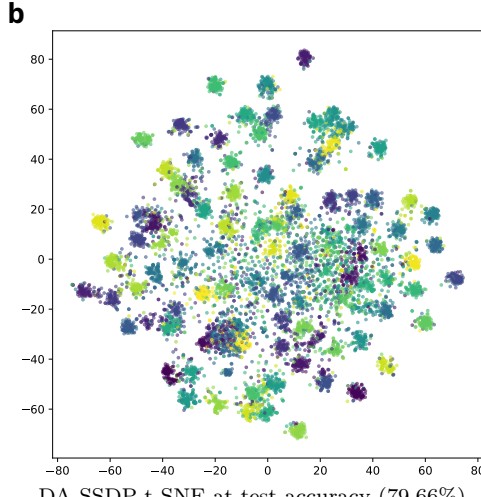

DA-SSDP t-SNE at test accuracy (79.66%).

Figure 7: (a) t-SNE visualization of hidden-layer embeddings after the final attention stage for the baseline model, exhibiting compact, well-separated clusters. (b) t-SNE visualization for the DA-SSDP model, exhibiting broader intra-class manifolds and smooth, low-density "bridges" that connect semantically related classes, indicating softer inter-class boundaries. Each dot is one sample after a two-dimensional t-SNE projection and its color is the class label. A cluster is a patch of the same-color dots that lie close together. The cluster area gauges how far samples of one class spread inside the feature space (small area = high tightness). The gap between clusters reflects the distance between class centers (large gap = clear separation).

## A.5 Hardware Perspective

### A.5.1 Neuromorphic hardware/ASIC implementation

Parts of DA-SSDP are hardware-friendly and biologically inspired in ways that align with neuromorphic/ASIC designs. Neuromodulators like dopamine are highly efficient in ASICs because they can be implemented as a simple multiplier or broadcast operation, rather than complex per-synapse computations. Neuromorphic hardware often supports global signals for modulation, making DA-SSDP more feasible for on-chip learning than a purely software-based regularizer. Its primary focus is on offline training, but it has components that could be adapted for hardware, emphasizing biological plausibility and efficiency.

Training needs batch statistics (synchrony $S_b$ and loss $\ell_b$), a dopamine-slope calibration, and operations such as outer products, exponential, and clipping. Supporting this would add extra compute and storage on chip (e.g., accumulators for $S_b$ and buffers for $\Delta t$). That increases design complexity, memory traffic, and power. The exact overhead depends on the architecture.

### A.5.2 Energy and spike statistics

In our results, DA-SSDP changes spike patterns by concentrating task-relevant activity into short, synchronous bursts. We observe a $33\times$ increase in median synchrony $S_b$ (Fig. 5). On event-driven hardware, power consumption often correlates with the number of synaptic events and the routing they require. If DA-SSDP lowers effective event counts in deployment, energy can go down. However, our software estimates are almost unchanged (e.g., 493 $\mu J$ *vs.* 494 $\mu J$ on CIFAR-100; Table 4) and so we do not make numeric energy claims. Any sparsity effect is task-dependent and should be checked by counting events directly.

### A.5.3 Platforms and feasibility evidence

Prior work on supervised deep SNN training and threshold regularization addresses activity balance and stability, and surveys describe neuromorphic ASICs and training pipelines that manage sparsity and noise (Sakemi et al., 2023; Lee et al., 2016; Bouvier et al., 2019; Hunsberger & Eliasmith, 2016; Pfeiffer & Pfeil, 2018; Evans et al., 2015). Bursty activity concentrates mainly on switching, so power delivery, clocking/asynchronous design choices, and thermal margins need extreme care. Storing spike latencies can use sparse address-event representation (AER) and buffer hierarchies tuned to the workload (Yamazaki et al., 2022). IBM TrueNorth demonstrated large-scale, event-driven inference with off-chip training (Merolla et al., 2014), while Intel Loihi supports programmable on-chip plasticity, including reward/three-factor mechanisms with evidence that neuromodulated learning rules are realizable in silicon (Davies et al., 2018). Memristive devices have also shown dopamine-like modulation of STDP windows at the device level, suggesting potential hybrid ASIC memristor pathways (Nikiruy et al., 2019). If the full DA-SSDP is too complex for direct hardware implementation, simplified versions using local-only rules can work on-chip.

### A.6 Theoretical Analysis: Complementary Information and Regularization

Let

$$\Delta \mathcal{L}^{(1)} = \eta \, \nabla_W \mathcal{L}_{\sup}(W) : \Delta W, \tag{15}$$

where $A : B \equiv \sum_{i,j} A_{i,j} B_{i,j}$ and $\eta > 0$ absorbs the scale in Eq. 1. Ignoring only the element–wise clipping (which clips magnitudes but not signs), write Eq. 1 as

$$\Delta W = \frac{1}{B} \sum_{b=1}^{B} G_b \, U_b, \qquad [U_b]_{i,j} = g_{b,i,j} \left( (A_+ + A_-) \lambda_{b,i,j} - A_- \right), \tag{16}$$

where $g_{b,i,j} \equiv g(\Delta t_{b,i,j})$ and $\lambda_{b,i,j} = Q_{b,i} P_{b,j}$ are exactly those in Eqs. 1–4. Using the linear (unclipped) form of the gate in Eq. 10,

$$G_b = 1 + k\hat{S}_b, \qquad \hat{S}_b = \frac{S_b - \mu_S}{\sigma_S}. \tag{17}$$

Then we have the exact identity

$$\mathbb{E}\left[\Delta \mathcal{L}^{(1)}\right] = \eta \mathbb{E}\left[\nabla_W \mathcal{L}_{\sup}(W) : \bar{U}\right] + \eta \, k \mathbb{E}\left[\hat{S}_b \left(\nabla_W \mathcal{L}_{\sup}(W) : U_b\right)\right], \qquad \bar{U} \equiv \frac{1}{B} \sum_b U_b. \tag{18}$$

In the implementation, the gate entering Eq. 1 is the clipped quantity $\mathrm{clip}(1 + k\hat{S}_b, 0, 2)$. Using the linear gate in Eq. 17 only enlarges magnitudes; therefore, any upper bound obtained from Eq. 18 continues to hold (conservatively) after clipping. For any $z \in \mathbb{R}$,

$$\left| \mathrm{clip}(1 + kz, 0, 2) - 1 \right| \leq |kz|. \tag{19}$$

There exists $\beta > 0$ such that, conditional on $S_b$,

$$\mathbb{E}\left[\nabla_W \mathcal{L}_{\sup}(W) : U_b | S_b\right] \leq -\beta \left(S_b - \mu_S\right). \tag{20}$$

Equivalently, by the law of total expectation and $\hat{S}_b$,

$$\mathbb{E}\left[\hat{S}_b \left(\nabla_W \mathcal{L}_{\sup}(W) : U_b\right)\right] = \frac{1}{\sigma_S} \mathbb{E}\left[(S_b - \mu_S)\left(\nabla_W \mathcal{L}_{\sup}(W) : U_b\right)\right] \leq -\beta \sigma_S. \tag{21}$$

Combining this assumption with Eq. 18 yields

$$\mathbb{E}\left[\Delta \mathcal{L}^{(1)}\right] \leq \eta \mathbb{E}\left[\nabla_W \mathcal{L}_{\sup}(W) : \bar{U}\right] - \eta \, k \beta \sigma_S. \tag{22}$$

Hence:

- **Informative synchrony (complementary information).** If $\text{corr}(S, \ell) < 0$ in warm–up, then $k > 0$ by Eq. 9, and the gated component contributes a *strictly negative* first–order term $-\eta k \beta \sigma_S$ in Eq. 22. The net sign depends on the baseline term $\mathbb{E}[\nabla_W \mathcal{L}_{\text{sup}} : \bar{U}]$, but the *increment due to gating* is negative and proportional to $k \beta \sigma_S$.

- **Weak synchrony (regularizer).** If $\text{corr}(S, \ell) \approx 0$ so that $k \approx 0$, the gated contribution in Eq. 18 vanishes to first order and the update reduces, in expectation, to the loss–agnostic baseline $\bar{U}$. With element–wise clipping in Eq. 1, this baseline acts as a *bounded regularizer* shaping co–activation statistics; we do not assume a favorable sign for $\mathbb{E}[\nabla_W \mathcal{L}_{\text{sup}} : \bar{U}]$. This baseline does not depend on $\ell_b$ and remains loss-agnostic by construction.

## A.7 Complexity of timing logs and updates

Consider a hooked module with $C_{\text{in}}$ inputs, $C_{\text{out}}$ outputs, mini-batch size $B$, and window length $T$. We denote the average spikes per sample per channel by $r_{\text{pre}}$ (input) and $r_{\text{post}}$ (output). Let $t_{\text{bytes}}$ be the bytes used to store one spike time (4 for FP32, 2 for FP16), and $b_{\text{bool}} = 1$ byte for a Boolean.

**DA-SSDP (ours).** From the implementation, we cache only the first spike time per channel $t^{\text{pre}}, t^{\text{post}}$ and form $\lambda_{b,i,j} = Q_{b,i} P_{b,j}$ and $\Delta t_{b,i,j} = |t_{b,i}^{\text{post}} - t_{b,j}^{\text{pre}}|$ by broadcasting. The extra memory is

$$\mathcal{M}_{\text{DA-SSDP}} = B(C_{\text{in}} + C_{\text{out}}) t_{\text{bytes}}. \tag{23}$$

The per-batch time consists of extracting first spikes (no pairwise loop)

$$\mathcal{T}_{\text{first}} = O\big(BT(C_{\text{in}} + C_{\text{out}})\big), \tag{24}$$

and forming the update tensor once by outer product and broadcasting

$$\mathcal{T}_{\text{update}} = O\big(B\, C_{\text{out}} C_{\text{in}}\big). \tag{25}$$

Warm-up statistics $(\mu_S, \sigma_S, k)$ use two scalars per batch and can be streamed with $O(1)$ memory.

**Conventional local rules for comparison.** We contrast three usual variants:

(i) Full multi-spike logging (pairwise STDP): store all spike times to reconstruct latencies.

$$\mathcal{M}_{\text{log}}^{\text{multi}} = B\Big[C_{\text{in}}(r_{\text{pre}}\, t_{\text{bytes}}) + C_{\text{out}}(r_{\text{post}}\, t_{\text{bytes}})\Big], \qquad \mathcal{T}_{\text{pair}} = O\big(BTC_{\text{out}} C_{\text{in}}\big). \tag{26}$$

(ii) Neuron traces (factorized STDP): maintain per-neuron exponential traces and update with $Q_t P_t^\top$ at every step $t$. This avoids storing all spike times but still does a pairwise outer product per step,

$$\mathcal{M}_{\text{trace}}^{\text{neuron}} = O\big(B(C_{\text{in}} + C_{\text{out}})\big), \qquad \mathcal{T}_{\text{trace}} = O\big(BTC_{\text{out}} C_{\text{in}}\big). \tag{27}$$

(iii) Eligibility traces per synapse (e-prop style): keep an eligibility state for each $(i, j)$.

$$\mathcal{M}_{\text{elig}}^{\text{syn}} = O\big(B\, C_{\text{out}} C_{\text{in}}\big), \qquad \mathcal{T}_{\text{elig}} = O\big(B\, T\, C_{\text{out}} C_{\text{in}}\big). \tag{28}$$

Combining Eqs. 23–28, DA-SSDP replaces either $O(BT(C_{\text{in}} + C_{\text{out}}))$ storage (full traces) or $O(BC_{\text{out}} C_{\text{in}})$ state (per-synapse eligibility) with a constant-size first-spike cache $O(B\,(C_{\text{in}} + C_{\text{out}}))$, and removes the $T$-factor from the pairwise update cost:

$$\text{DA-SSDP: } O\big(BT(C_{\text{in}} + C_{\text{out}}) + BC_{\text{out}} C_{\text{in}}\big) \quad \text{vs.} \quad \text{STDP/e-prop: } O\big(BTC_{\text{out}} C_{\text{in}}\big). \tag{29}$$

For deep modules where $C_{\text{out}} C_{\text{in}}$ dominates and for longer windows $T$, the saving is substantial in both memory and time.

---

**Algorithm 1** DA-SSDP (post-step correction; used in our experiments)

---

**Require:** Mini-batch $\{(P_b, Q_b, t_b^{\mathrm{pre}}, t_b^{\mathrm{post}}, \ell_b)\}_{b=1}^B$; hooked weight $W \in \mathbb{R}^{C_{\mathrm{out}} \times C_{\mathrm{in}}}$; $A_+, A_-, \sigma > 0$; warm-up $E_{\mathrm{warm}}$; statistics $(\mu_S, \sigma_S, k)$ fitted after warm-up.

1: **Supervised step:** forward $\to$ loss $\ell = \frac{1}{B} \sum_b \ell_b \to$ backprop $\to$ `optimizer.step()`.
2: **Warm-up:** compute per-sample $S_b$.
3: **if** epoch $< E_{\mathrm{warm}}$ **then** record $(S_b, \ell_b)$ and **return**
4: **end if**
5: **if** epoch $= E_{\mathrm{warm}}$ and $(\mu_S, \sigma_S, k)$ not fitted **then** standardize $S_b, \ell_b$; fit $k$.
6: **end if**
7: **for** $b = 1$ **to** $B$ **do**
8:      $G_b \leftarrow \mathrm{clip}\big(1 + k\,(S_b - \mu_S)/\sigma_S, 0, 2\big)$
9: **end for**
10: For each $(i, j)$, set $[U_b]_{i,j} \leftarrow g_{b,i,j}\big((A_+ + A_-)\,\lambda_{b,i,j} - A_-\big)$
11: $\Delta W \leftarrow \mathrm{clip}\big(\frac{1}{B} \sum_{b=1}^B G_b U_b, -1, 1\big)$
12: **Post-step correction:** $W \leftarrow W + \Delta W$                    decoupled from optimizer moments

---

## A.8 Algorithmic Specification

Algorithm 1 specifies the post-step integration used in our experiments. After the standard supervised update, DA-SSDP computes the per-sample synchrony $S_b$, the per-sample gates $G_b$, and the local matrices $U_b$. The aggregated, element-wise clipped correction $\Delta W$ is then applied. Because this correction is added after the optimizer step, it is decoupled from optimizer state: Adam/AdamW's first and second moments continue to track only the supervised gradient, while the DA-SSDP correction remains a small, bounded parameter adjustment. This decoupling is precisely why the method is optimizer-agnostic. For completeness, we note that one could alternatively inject the same local term into the gradient before the optimizer step if one wished adaptive moments to absorb it; however, we do not use that variant in this work.

## A.9 Supplementary Experiments

To assess optimizer compatibility and architectural generality, we integrated DA-SSDP into the DH-SRNN backbone Zheng et al. (2024) trained with Adam on SHD. DA-SSDP is attached to the classifier as a post-step correction. No hyperparameters were retuned.

**Result.**    On SHD, DH-SRNN with Adam reaches 88.1% accuracy; adding DA-SSDP at the classifier improves the average accuracy to 89.3% (+1.2%).

Table 5: DH-SRNN on SHD with Adam.

| Method | Optimizer | Acc. (%) |
|---|---|---|
| DH-SRNN (baseline) | Adam | 88.1 |
| + DA-SSDP | Adam | 89.3 |

The improvement with Adam and a non-transformer backbone indicates that DA-SSDP is optimizer-agnostic and transfers beyond SpikingResformer.

## A.10 Declaration of Generative AI

The authors used LLMs in some parts of the paper in order to check grammar mistakes.

