# OpenReview forum: "Synchrony-Gated Plasticity with Dopamine Modulation for Spiking Neural Networks"
_TMLR — Accepted by TMLR_

### Review · Reviewer_728F · 2025-10-15

**Summary Of Contributions:**

This study targets improving spiking neural networks. The authors proposed a post-update rule, which applies an update on weight using a Gaussian kernel with a dopamine gate. Experiments such as CIFAR and ImageNet datasets verify the improvements. Several qualitative analyses are presented in the appendix.

**Audience:**

Yes

**Audience Explanation:**

Researchers on spiking neural networks would find certain values in this study.

**Claims And Evidence:**

No

**Claims Explanation:**

I think current version is not sufficient. Please see the Requested Changes below.

**Requested Changes:**

- The proposed method applies update after backpropagation. This practice is only compatible with stochastic gradient descent; for example, the Adam(W) optimizer uses the second moment and does not directly apply gradients on parameters.

- Eq. 9 computes the outer product with division by channel sizes, but mini-batch size is not considered and might affect the result. The authors should clarify the definition as per-sample or per-batch.

- “This difficulty stems primarily from the high memory demands of maintaining accurate spike-timing logs and the potential for purely local plasticity adjustments to clash with the supervised learning goal. To effectively leverage local signals derived from spiking neuron dynamics, …” This part requires clear empirical or theoretical evidence. The authors should quantify and measure this behavior more concretely.

- I think that 79% accuracy on ImageNet-1K is a solid baseline, and achieving improvements on this baseline is impressive. The ablation study was also conducted faithfully, and I think that the overall experimental results are promising. One comment is that the authors should consider other models beyond SpikingResformer.

- Nevertheless, this manuscript lacks theoretical analysis. I encourage the authors to perform and formulate some theoretical results.

- Please avoid excessive spacing management such as the narrow space between “clip(” in Eq. 14. Avoid writing in parentheses, such as the two parentheses in the Robustness paragraph and Section 4.3.1. Avoid excessive italics, such as in the Per-dataset hyperparameters paragraph. Avoid “and/or” and write precisely. Honestly speaking, I feel that several parts across this paper were written by GPT. Although it is not bad to be helped by GPT, I encourage the authors to clarify the use of LLM, at least in the appendix.

- Writing should be improved.
    - Check parenthesis for “Conventional regularizers (e.g., weight decay …” on page 2.
    - “as showed” → “as shown”
    - CriterionWarpper → CriterionWrapper
    - “we fall back to a neural gate” → I think it should be a neutral gate, right?
    - “making population activity becomes more coordinated and stable.” → “making population activity more coordinated and stable.”
    - There is a duplicate entry for the paper “Spike-driven transformer.”
    - “Hebb (2005)” → “Hebb (1949)”
    - Please use \citep and \citet separately.

---

> ### Author Response · Authors · 2025-11-04
> **Response to Reviewer 728F.**
>
> We would like to thank you for your professional review work, constructive comments, and valuable suggestions on our manuscript. Your agreement with our conclusions and results has been a powerful assessment of the validity of our research.
>
> **For Point 1 (post-step update; “only compatible with SGD”) & Point 4 (try other architectures):**
>
> We thank the reviewer for raising these important points. DA-SSDP is applied as a small, bounded post-step parameter correction, so it does not rely on any SGD-specific property. To address both optimizer compatibility and architecture generality, we additionally integrated DA-SSDP into a different model, DH-SRNN(https://www.nature.com/articles/s41467-023-44614-z), trained with **Adam** on **SHD**. Hooked at the classifier, DA-SSDP improved accuracy by +1.2 % over the DH-SRNN baseline (**88.1% → 89.3%** on average). This shows that DA-SSDP transfers beyond spiking transformers and does not depend on the choice of optimizer.
>
>
> **For Point 2 (Eq. 9: per-sample vs per-batch; effect of mini-batch size):**
>
> We appreciate the opportunity to provide further clarification on this important point. Eq. 9 defines a **per-sample** synchrony score $S_{b}$ : for each sample $b$, we take the outer product of that sample’s spike indicators and normalize only by the channel counts $C_{out}C_{in}$. We intentionally do not divide by the mini-batch size $B$, because $S_{b}$ is a per-sample statistic used to compute a per-sample gate $G_{b}$. The weight update then averages over samples through the $1/B$ factor in Eq. 5, so changing $B$ affects estimator variance but not the update’s scale. We will revise the text around Eq. 9 to state explicitly that $S_{b}$ is per-sample and to note its independence from $B$.
>
>
> **For Point 3 (evidence for timing-log burden and the “potential clash” of purely local plasticity):**
>
> We thank the reviewer for bringing this to our attention.
> On the “potential clash”: We do not claim that purely local plasticity necessarily conflicts with the supervised objective. we clarify that, because a purely local update is computed without the supervised gradient, its contribution is not sign-guaranteed. Appendix A.6 now makes this precise: the update decomposes into a loss-agnostic baseline and a gated part whose sign is determined by data-measurable synchrony. When synchrony is informative, the gated part makes a strictly descent-aligned contribution; when synchrony is weak, the gate becomes neutral and the update reduces to a bounded regularizer.
>
> On memory/compute: The new appendix gives a reproducible comparison. In brief, DA-SSDP caches only the first spike per channel and forms co-firing once per batch, removing the time-window loop; in contrast, multi-spike STDP, neuron-trace STDP, and per-synapse eligibility methods either process every time step across all pre/post pairs or maintain per-synapse state. Full scalings and derivations are in the appendix.
>
> **For Point 5 (theoretical analysis):**
>
> Based on the reviewer’s suggestion, we added Appendix “Theoretical Analysis: Complementary Information and Regularization.” To offer more theoretical analysis. In this part, we analyzed: (i) informative synchrony acts as complementary information, and (ii) weak synchrony collapses the gate to neutral, then DA-SSDP works as a bounded regularizer.
>
> **For Point 6 (formatting, style, and LLM):**
>
> Thank you to the reviewer for the detailed suggestions and for pointing out our oversight. Following these comments, we made the requested edits throughout: removed tight spacing, reduced italics, and replaced “and/or” with precise wording. We also corrected minor issues (e.g., citation macros, typos) and added a claim in the appendix clarifying any language-editing assistance and LLMs. We also fixed the \citep and \citet issues based on the advice.
>
> **For the writing problems:**
> Once again, we thank the reviewer for the careful reading and thoughtful corrections. We have implemented all of the suggested changes and revised the manuscript accordingly.

---

> > ### Comment · Reviewer_728F · 2025-11-06
> > **Thank you for your response**
> >
> > Thank you for your response.
> >
> > - Regarding point 1, however, I am still unsure about how the first and second moments in Adam can incorporate the post-step update of the proposed method. I would like to request to explicitly state algorithm pseudocode in the main text.
> >
> > - Regarding point 4, I appreciate the additional results with DH-SRNN, but I found that the revised manuscript does not include this result. Please check this point.
> >
> > I believe other parts have been successfully revised.

---

> > > ### Author Response · Authors · 2025-11-18
> > > **Response to Reviewer 728F (follow-up)**
> > >
> > > Thank you for the follow-up and further suggestion.
> > >
> > > **For point 1 (Adam moments and pseudocode in the main text):**
> > >
> > > We now state the DA-SSDP procedure explicitly in Appendix A.8 Algorithmic Specification.
> > >
> > > **For point 4 (including the DH-SRNN result):**
> > >
> > > We have included the additional experiment in the revised manuscript A.9 Supplementary Experiments.
> > >
> > > We hope these changes address your remaining concerns, and we appreciate your helpful guidance.

---

### Review · Reviewer_nwUT · 2025-10-23

**Summary Of Contributions:**

Authors propose a special training (or regularisation) strategy for spiking neural networks. In addition to surrogate backpropagation they apply an update based on neuron synchrony. Given that the update rule is also related to Hebbian plasticity, authors made an analogy with synaptic modifications produced by dopamine release. The resulting method is called DA-SSDP.

Authors apply DA-SSDP in two variants for the training of SpikingResformer. On four classification problems DA-SSDP led to the improved accuracy.

**Audience:**

Yes

**Audience Explanation:**

Researchers in spiking neural networks and other biologically inspired approaches may be interested in the current contribution.

**Claims And Evidence:**

No

**Claims Explanation:**

Authors claim several contributions:
1. Introduction of the synchrony-based method.
2. Scalable implementation of spiking transformer.
3. Validation that shows DA-SSDP provides consistent accuracy improvements.

I am uncertain about the correctness of the second and the third contributions.

**Requested Changes:**

I have several questions about a plasticity mechanism that authors propose in the paper. My main concern is that I can not understand what kind of problem authors are trying to solve with DA-SSDP.

**Scalable implementation of SNN-Transformer.**

On pages 2 and 3 authors claim "(B) Memory-efficient, scalable implementation for deep SNN-Transformers: DA-SSDP requires only binary spike indicators and first-spike latencies (...) We provide a drop-in implementation that tweaks just the final classification part and a simple late-stage feature projector, keeping the architecture and training schedule (epochs, batch size, optimizer) unchanged. This approach saves a lot of memory compared with storing detailed spike-timing records."

The title of the contribution (given in bold font in the text of the article) suggests that authors provide a scalable implementation for deep SNN-Transformers. Is this the case? If I am not mistaken, authors used SpikingResformer https://arxiv.org/abs/2403.14302 which they altered in several places by introducing DA-SSDP plasticity applied only on the training stage. Moreover, authors claim that their approach keeps "all forward-pass operations intact" (page 5). If this is the case, the implementation of SpikingResformer remains largely the same. Can the authors please clarify what they mean by scalable implementation of deep SNN-Trainsormers?

**Consistent accuracy improvements.**

Accuracies of SpikingResformer, SpikingResformer+SSDP, SpikingResformer+DA-SSDP are within . It seems that the improvement is within $2\sigma$. Is it "statistically significant"? For example, if one applies a different train-test split, does the difference in performance survive? Can the authors provide accuracy with $\pm 2\sigma$ "confidence intervals"?

**Many components of proposed approach are not biologically plausible:**

1. To define plasticity authors use batch size.
2. There is a warmup state where rules are different from the rest of training.
3. There are train and deployment stages.
4. Gradient descent is used for training.
5. $G_b$ is computed using global information (this is acknowledged in limitations).
Can the authors comment on the biological plausibility of their approach? Is it important to have biological plausibility if surrogate backpropagation is used to train a spiking neural network?

---

> ### Author Response · Authors · 2025-11-04
> **Response to Reviewer nwUT.**
>
> We thank the reviewer for the sincere feedback, which has prompted us to re-evaluate and strengthen our arguments.
>
>
> **For Point 1 (Scalable implementation of SNN-Transformers):**
>
> Thank you to the reviewer for asking us to clarify what we mean by “scalable.” In our paper, “scalable” refers to the training-time behavior of the synchrony update, not to a new forward or inference implementation of SpikingResformer. It is **scalable with depth** because the extra work per attached is $O(BC_{\mathrm{out}}C_{\mathrm{in}})$ and uses only binary spikes and first-spike latencies, so it is almost independent of the number of time steps $T$ and adds negligible training time/energy even when used on several layers of deeper models. For more details about time complexity, please see Appendix 7. It is also **scalable across architectures** because the same update applies to any linear mapping that exposes pre/post spike events and first-spike latencies, without altering the backbone or its training schedule. For this we additionally integrated DA-SSDP into a different model, DH-SRNN(https://www.nature.com/articles/s41467-023-44614-z), trained with Adam on SHD. Hooked at the classifier, DA-SSDP improved accuracy by +1.2 % over the DH-SRNN baseline (88.1% → 89.3% on average). To avoid confusion, we have revised contribution (B) to “Scalable training-time synchrony update for deep SNN-Transformers,”.
>
> **For Point 2 (Consistent accuracy improvements):**
>
> The reviewer’s questions regarding our results have been thought-provoking, leading to a more rigorous examination of our results. We understand the concern about dataset settings. However, all our benchmarks use the standard, fixed public test sets (CIFAR-10/100 and ImageNet-1K), so changing the train–test split is not applicable. To assess robustness, we run every configuration five times with different random seeds (affecting data order, initialization, and mixup) and report mean ± std; on CIFAR-100 and ImageNet-1K the gains from DA-SSDP are statistically significant (p < 0.05) and the ordering SpikingResformer < SpikingResformer+SSDP < SpikingResformer+DA-SSDP holds across all seeds, while on CIFAR10-DVS the gain is small and not significant. Consistent with this, Section 4 Experiments (first paragraph) already states that each experiment is repeated five times for statistical reliability, and Section 4.2 discusses why improvements on CIFAR10-DVS are close to neutral in our setting. In the revision, Table 1 now reports five-run mean ± std (replacing single best numbers).
>
> **For Point 3 ( biologically plausible):**
> We appreciate the reviewer’s identification of our study’s limitations, which we have addressed with additional discussion and future work proposals. We clarify that our aim is not a biophysically faithful model but a training-time, biologically inspired regularizer that combines a local Hebbian synchrony signal with a dopamine-like global gate. Using batches serves as a practical short-trial estimate of population synchrony, which is needed when one needs stable statistics without maintaining per-synapse temporal traces; this choice is engineering-driven, and we state it clearly. The short warm-up calibrates the global gate once by measuring how synchrony correlates with task improvement, then the gate is fixed; this mimics modulatory gain setting or meta-plasticity rather than an ongoing rule change. The distinction between training and deployment reflects our goal of applying learning signals while keeping the forward dynamics unchanged at inference. We do use gradient descent to optimize the supervised objective, and the synchrony update is designed to be compatible with it, in the spirit of three-factor rules where a global neuromodulator scales local co-activity. The gate Gb is indeed global, and we acknowledge this limitation in the paper and frame it as a neuromodulatory broadcast that gates plasticity strength rather than a synapse-local quantity. To address plausibility concerns directly, we also report a two-factor variant without the global gate (i.e., fully local SSDP), which improves accuracy more modestly but in the same direction, while DA-SSDP yields the strongest and most stable gains. In short, we mean to indicate that we make a trade-off between biological inspiration and performance. Given that surrogate backpropagation is already used for the task objective, our contribution is a biologically motivated, training-time regularizer that measurably improves accuracy and introduces spike synchrony as a learning signal for SNNs.

---

> > ### Comment · Reviewer_nwUT · 2025-11-11
> >
> > I would like to thank the authors for providing clarifications.
> >
> > One of my main concerns was the absence of error bars in the reported top-1 accuracies. The reviewer 6cPD raised a similar point. I believe that the authors mostly resolved the issue by providing standard confidence intervals. Note, that there is a possibility to change train-test split for CIFAR-10 and ImageNet https://arxiv.org/abs/1902.10811, but it is a less standard practice so I do not insist authors should follow it.
> >
> > Authors also addressed two minor concerns regarding biological plausibility and the meaning of scalability of the approach. Regarding scalability, it is my understanding that authors claim their approach requires less memory and computation in comparison with naive synchrony detection that requires keeping a detailed spike-timing record. Can the authors please clarify how this alternative inefficient method can be implemented and provide an estimation of its memory requirement?

---

> > > ### Author Response · Authors · 2025-11-18
> > > **Response to Reviewer nwUT (follow-up)**
> > >
> > > We thank the reviewer for the additional clarifications and for pointing out the possibility of using alternative train–test splits for CIFAR-10 and ImageNet. In the current work, we keep the standard official splits to remain comparable with prior SNN and ANN baselines, but we agree that exploring different splits is an interesting direction.
> > >
> > > Regarding the question of “how this alternative inefficient method can be implemented and provide an estimation of its memory requirement”:
> > >
> > > Concretely, for a layer with $C_{\mathrm{in}}$ input and $C_{\mathrm{out}}$ output channels, batch size $B$ and sequence length $T$:
> > >
> > > A full spike-time logs implementation would, for every time step $t$, store the spike state for all neurons and then evaluate all pre/post pairs. This leads to either
> > > (i) neuron-trace storage of size $\mathcal{O}(B T C_{\mathrm{in}}) + \mathcal{O}(B T C_{\mathrm{out}})$, followed by an $\mathcal{O}(B T C_{\mathrm{in}} C_{\mathrm{out}})$ scan to find all coincidences, or
> > > (ii) an explicit per-synapse log of coincidences of size $\mathcal{O}(B T C_{\mathrm{in}} C_{\mathrm{out}})$ in the worst case.
> > >
> > > For a typical ImageNet-scale configuration in our experiments (for example $B{=}256$, $T{=}4$, $C_{\mathrm{in}}{=}C_{\mathrm{out}}{=}512$), this corresponds to on the order of a gigabyte of spike-log memory per layer if stored in 32-bit format.
> > >
> > > In contrast, our DA-SSDP implementation stores only
> > > (i) one binary “ever fired” indicator per channel,  $P_b\in\lbrace 0,1\rbrace^{C_{\mathrm{in}}}$ and $Q_b\in\lbrace 0,1\rbrace^{C_{\mathrm{out}}}$, and
> > >
> > > (ii) one first-spike time per channel, $t_b^{\text{pre}} \in \mathbb{R}^{C_{\mathrm{in}}}$ and $t_b^{\text{post}} \in \mathbb{R}^{C_{\mathrm{out}}}$.
> > >
> > > The pairwise latencies $\Delta t_{b,i,j}$ and synchrony mask $\lambda_{b,i,j}$ are then formed by broadcasting these $(C_{\mathrm{in}}{+}C_{\mathrm{out}})$ vectors over a single window, without keeping any per-synapse time series across $T$. Thus the extra memory for spike logs scales as $\mathcal{O}(B(C_{\mathrm{in}}{+}C_{\mathrm{out}}))$; in the same numeric example above this is on the order of megabytes rather than gigabytes.

---

### Review · Reviewer_6cPD · 2025-10-25

**Summary Of Contributions:**

### Summary
Spiking transformers have been mostly trained using surrogate backpropagation. The paper proposes a lightweight local update rule, DA-SSDP, that can complement surrogate backpropagation in the training of spiking transformers. Different from spike-timing-dependent plasticity (STDP), which induces much overhead when tracking spike times/traces, the proposed update rule depends on co-firing synchrony and only requires binary spike indicators and first-spike latencies, thus being more memory efficient than STDP, enabling its application in large-scale deep SNNs. This paper also incorporates dopamine modulation to align the local updates with the supervised learning signal. The paper demonstrates the utility of the proposed rule in image classification tasks.


### Strengths
- The proposed rule is lightweight, easy to implement, and induces little overhead during training.
- The proposed update rule could spark interest in synchrony-based learning rules, which are relatively under-explored.
- Extensive analyses of the proposed rule are performed

### Weaknesses
1. Statistics like confidence interval/standard deviation are not provided for experimental results.
2. Writing/Polish can be significantly improved
  - a) Structure of Introduction can be improved
    - The details of the proposed method are introduced in the second half of the third paragraph but the proposed method is only introduced later in the fifth paragraph.
  - b) Structure of Experiments can be improved
    - Figure 5 is described in Section 4.2, which is supposed to discuss the results in Table 1. Figure 5 itself is discussed later in much more detail in Section 4.4, so why include it in Section 4.2.
    - The results are discussed in Section 4.2 but the baselines and metrics are only introduced later in Section 4.3. Also, more discussion on the choice of the baselines (and perhaps metrics) is appreciated. For example, some sentence like: "We mainly compare with SpikingResformer (Shi et al, 2024), but performances of earlier SSN transformer variants \[references] are also included for reference."
  - c) No Background section
    - Section 3.1 and the description of the SpikingResformer architecture better belong to a separate Background section. Including them in the Methods section gives the impression that they are part of this paper's contribution.
  - d) Lack of consistency
    - Equations 6 and 8 are supposed to be components of Equation 5 but use slightly different notations, e.g., $g_{b, i, j}$ vs $g(\Delta t_{b, i, j})$
    - The paper sometimes writes $\lambda=0$ and sometimes $\lambda$=0. The part "=0" should also be included in the dollar signs for the equation. This happens a lot throughout the paper.
    - The paper sometimes capitalize "Transformer" and sometimes do not. Similar for "Classifier".
  - e) Lack of polish for figures
    - In Figure 3a, the color bar overlaps with the figure.
    - In Figure 4 (top), the horizontal axis ticks 0.5, 1.5, and 2.5 are not necessary. Plus the horizontal axis label is missing.
    - In Figure 5, why do you report the data divided by 0.0028 for the baseline?
  - f) The paper does not distinguish between the use of \citet and \citep.
  - g) Grammar
    - For example, in the second paragraph of Related Works (first letter of "works" should be capitalized by the way): "First, scalability by keeping super detailed records
of spike timings and assigning credit/blame between pairs of neurons over time, using up a huge amount of
memory and processing bandwidth" is not a complete sentence. Also, "super" is not a word formal enough to be used in research papers.
    - Another example, in the caption of Figure 5: "making population activity becomes more coordinated and stable"
  - h) References
    - "An image is worth 16 x 16 words" is published in ICLR 2021, so the paper should cite that version instead of the arXiv version. Also, the author list for this paper is wrong. Alexey Dosovitskiy is not the sole author of this paper.
    - For "Rethinking the inception architecture for computer vision", "recognition" of CVPR should be capitalized.
    - Why does "Spike-driven transformer" appear twice in the references, both published in NeurIPS but with a different year?
    - After "Advances in Neural Processing Systems", the paper sometimes include "(NeurIPS)" and sometimes do not.
    - I just did a brief scan of the references. There might be other issues.
  - i) the use of acronyms without first introducing the full term, e.g., LIF and STDP
  - j) In Section 4.3.2, There's a line consisting of only two words "Hyperparameter sensitivity". Is that a mistake?
3. The proposed method bears similarity to Gaussian Decay-based Weight Update Mechanism in the cited paper "Synchrony-Gated Plasticity with Dopamine Modulation for Spiking Neural Networks", but the connections are not discussed.

**Additional Comments:**

Clarifying Questions
1. Is $N=B E_{warm}$?
2. In Equation 5, when only one neuron of a pair spikes, the one that didn't spike has $t=T$, then $\Delta t$ is equal to the difference between time of the one that spiked and $T$. Is this intended? If yes, what's the intuition?
3. It seems that SSDP is introducing the inductive bias that, if two neurons co-fire together more often than not, then they should be encouraged to co-fire consistently across all batches of data. Similarly, if two neurons co-fire less often than not, their connections are weakened. Is this an accurate description? If yes, is there an explanation of why this should improve performance?
4. Is is possible for $k$ to be negative? Also, is it possible that after some updates, the correlation between synchrony score and supervised loss change? For example, originally high synchrony is correlated with low loss, but over time, due to updates by surrogate backprop, low synchrony becomes correlated with low loss. How would the fixed ($\mu_S, \sigma_S, k$) still be relevant?
5. Can you provide more explanation for Figure 3?
6. In Section 4.4, the paper writes "After introducing DA-SSDP, neurons crucial for decision become more prone to simultaneous activation and as a result, these channels are reinforced and maintained elevated spiking rates, while more neurons emit at most one spike or stay inactive." This is very hard to see from Figure 4. Wouldn't it be much easier to demonstrate the claim simply by comparing the histograms of spike counts of the baseline and DA-SSDP?
7. Is there a reason to break eq 15 into two lines?

**Audience:**

Yes

**Audience Explanation:**

A lightweight local update rule that is easy to implement and improves the performance of SNN transformers is of interest to researchers in this area.

**Claims And Evidence:**

Yes

**Claims Explanation:**

Claims are supported in general, with some concerns about experiment results.

- Claim 1 (bringing synchrony-based signal into model learning) is supported by the detailed description of the proposed learning rule in Section 3. Claim 2 (memory-efficient, scalable implementation for deep SSN-Transformers) is supported likewise.
- Claim 3 (quantitative validation and robust analysis with safe degradation) is supported by results in Section 4. However, there are several concerns:
  - The lack of metrics of statistical significance (e.g., confidence intervals or standard deviation) casts doubts on whether the improvements reported in Table 1 occurred by chance. The paper mentions running 5 trials for every experiment, so relevant statistics should be easy to compute and report.
  - Why is there no Prologue+DA-SSDP and Classifier+DA-SSDP in Table 2? It is strange that the paper includes the results for all variants apart from these two.
  - In the interpretation paragraph of Section 4.3.2, the paper writes that the dopamine gate "delivers a stable, batch-specific rescaling ... which compresses the per-batch update magnitude". But according to Eq. 14, isn't the range of the dopamine gate [0, 2]? So it does not only compress, but could increase the magnitude.
  - In Section 4.4, the paper writes "In the baseline network, spike frequencies vary broadly, and most neurons fire every time step (four spikes in total)." However, judging from the bottom part of Figure 4, very few neurons fire every time step (spike count = 4), seemingly contradicting with the statement.

**Requested Changes:**

Critical:
- Report statistical significance (e.g., confidence intervals) for Table 1

Highly recommended:
- Improve writing/polish (see Weaknesses above)

---

> ### Author Response · Authors · 2025-11-04
> **Response to Reviewer 6cPD (Part 1).**
>
> Thank you for your careful reading and thoughtful suggestions. We have carefully revised the paper’s writing and presentation based on your comments. Below we highlight several specific changes and clarifications that address your concerns:
>
> **For weakness (a), Introductory order:**
>
> We reordered the early exposition so that DA-SSDP is introduced first, followed by the spiking-neuron model and implementation details. This makes the statement more consistent.
>
> **For weakness (b), Placement of the Figure 5 discussion and citations in setup:**
>
> To avoid interrupting the results narrative, we removed the discussion of Figure 5 from Section 4.2 and added citations to other SNN-transformer variants.
>
> **For weakness (c), Background and Methods boundary:**
>
> We moved the SpikingResformer backbone out of Methods and into a concise Background paragraph, avoiding implying it is part of our contribution. Methods now focus solely on our synchrony update.
>
> **For weakness (e), Figure problems:**
>
> Values are normalized **for visualization only** by dividing $S_{b}$ by each method’s test-set 99.5th percentile computed over positive $S_{b}$; the right panel simply zooms into the top range. All reported statistics and significance tests use raw (unnormalized) $S_{b}$. We also polished figures based on your suggestions.
>
> **For weakness 3, similarity to cited paper:**
>
> We added more explanation in the introduction part to better explain the difference between this paper and the cited paper.
>
> **Responses to specific questions:**
>
> We thank you again for your careful check on writing issues as well as polish issues. We fixed them based on your comments.
> For the third concern in Claim3, we change the statement to ‘which **adjusts** the per-batch update magnitude’; and about the statement related to Fig.4, we change it to ‘**and a relatively large part of neurons** fire every time step (four spikes in total)’.
>
> **1.Statistical reporting and more ablation studies:**
>
> We now report mean ± standard deviation throughout (see Table 1). In addition, Prologue+DA-SSDP and Classifier+DA-SSDP have been added to Table 2 for completeness.
>
> **2.Clarifying questions:**
>
> **(i) For question 1:**
>
> $N$ denotes the number of warm-up mini-batches.
>
> **(ii)For question 2:**
>
> we think of $T$ as the decision deadline and we want the strongest depression to fall on pairs that are still out of synchrony gate near this deadline, while avoiding over-penalizing very early one-sided spikes. When only one neuron fires, the pair is treated as asynchronous and the update goes to depression. If the other neuron fired very early (e.g., $t\approx 0$), this is an isolated early blip whose effect on the final decision is uncertain, so we keep the penalty small. If the other neuron fires only near the end of the window ($t\to T$), then the pair remains unsynchronized at the readout horizon; this is more likely to interfere with downstream decisions, so we depress that connection more strongly. The rule therefore makes the network align useful co-activation with the decision period rather than reacting to early, one-off events. If neither neuron spikes, we set $\Delta t=0$, so the strongest depression term remains, penalizing connections that never co-activate.
>
> **(iii) For question 3:**
>
> DA-SSDP introduces a bias that strengthens pairs that repeatedly co-fire within the temporal window and weakens pairs that seldom do. With the loss-aware gate, this bias is aligned with the supervised objective, encouraging class-consistent co-activation, reducing asynchronous cross-talk, and yielding more stable, more linearly separable representations.
>
> **(iv) For question 4:**
>
> By construction, $k=-\mathrm{corr}(\hat S,\hat\ell)$ is estimated once during warm-up, so $k$ can be positive or negative. A negative $k$ simply down-weights high-synchrony batches; it does not flip update signs because $G_b$ only rescales the local SSDP term and is clipped to $[0,2]$. Regarding drift, two points matter. First, we standardize $S_b$ with the warm-up $(\mu_S,\sigma_S)$ and then clip $G_b$, so later scale shifts in synchrony cannot induce large or unstable modulation. Second, DA-SSDP is applied after surrogate backpropagation and has a small magnitude compared to the supervised gradient. Therefore, a late reversal of the synchrony–loss correlation cannot dominate the learning signal. In short, the fixed $(\mu_S,\sigma_S,k)$ amplifies synchrony only when early evidence indicates benefit, and otherwise safely defaults toward neutrality due to standardization and clipping.
>
> **3. Fig. 3 explanation:**
>
> We expanded the discussion of Fig. 3 in Section 4.4.
>
> Due to the character limit, the remaining part of our response continues in the next comment titled “Response to Reviewer 6cPD (Part 2)”.

---

> > ### Author Response · Authors · 2025-11-04
> > **Response to Reviewer 6cPD (Part 2)**
> >
> > This comment continues our response to Reviewer 6cPD (Part 2) and addresses the remaining points.
> >
> > **4. Why raster plots rather than spike-count histograms?**
> > Thank you for this suggestion. we clarify that this figure concerns time-locked synchrony, not only rate changes. Spike-count histograms summarize per-neuron totals and discard timing, synchronous bursts and dispersed spikes can therefore appear identical. We mean to indicate that raster plots preserve timing and expose synchrony as vertical alignments, directly evidencing the phenomenon under study.
> >
> > **5. Equation formatting:**
> >
> > We corrected Eq. (15) to a single-line expression; the previous line break was a formatting artifact and did not affect the definition.

---

> > > ### Comment · Reviewer_6cPD · 2025-11-11
> > > **Thank you for your response**
> > >
> > > Thank you for your response. My concerns are largely addressed.
> > >
> > > Some remaining concerns:
> > > 1. In my review, weakness 2a refers to Section 1 (Introduction), not Section 3 (Methods). I was thinking that there was too much exposition on the details of your method, before you actually introduced your method by "In this work, we present a novel training strategy...".
> > >
> > > 2. Regarding weakness 2c, I was thinking of having a separate section rather than just moving them to the end of Section 3, but Section 3.3 looks good where it is right now. I'm not sure if you need Section 3.2 though. It does not seem like the LIF mechanism is used anywhere else in the paper. Finally, if you change the order of the sub-sections in Section 3, remember to also change the overview of the section. The sentence "We begin by outlining the transformer architecture used in our experiments, followed by a detailed explanation of the DA-SSDP rule" is now inaccurate.
> > >
> > > 3. There should not be $(\Delta t_{b, i, j})$ in Eq. 1 if you're aligning notations.
> > >
> > > 4. Is there no way to adjust Fig 3a so that the color bar does not overlap with the images, like in Fig 3b?
> > >
> > > 5. I still don't understand why from Fig 4, you say that "a relatively large part of neurons fire every time step" for the baseline model. I might be misunderstanding things here, but if a neuron fires every time step, shouldn't the spike count be 4 at the bottom of Fig 4? But when we look at the blue plot at the bottom of Fig. 4, only a few neurons, I suppose $<50$, of the 500 neurons have a spike count of 4.

---

> > > > ### Author Response · Authors · 2025-11-18
> > > > **Response to Reviewer 6cPD (follow-up)**
> > > >
> > > > **For question 1:**
> > > >
> > > > Thank you for the clarification that weakness 2a was meant to refer to Section 1. We have revised the Introduction to keep it focused on motivation and high-level goals. Detailed exposition of DA-SSDP is now confined to the Methods section and the contribution summary after that sentence.
> > > >
> > > > **For question 2:**
> > > >
> > > > Thank you for the clarification regarding weakness 2c. In the revised version, we removed the previous part related to LIF neuron dynamics, which is unnecessary for our method. We also updated the overview sentence at the beginning of Section 3 so that it now matches the actual order: we first describe the DA-SSDP rule and then explain how it is integrated into the SpikingResformer backbone and trained in two stages.
> > > >
> > > > **For question 3:**
> > > >
> > > > We thank the reviewer for pointing out the notational inconsistency in Eq.1.
> > > >
> > > > We now treat $g_{b,i,j}$ as the kernel value and define
> > > > $g_{b,i,j} = \exp[-\Delta t_{b,i,j}^{2}/(2\sigma^{2})]$, removing the redundant argument
> > > > $(\Delta t_{b,i,j})$ in Eq.1 so that the notation is aligned.
> > > >
> > > > **For question 4:**
> > > >
> > > > Thank you for pointing this out. We agree that the layout of Fig.3(a) could be improved. However, regenerating this panel with a new layout would require re-running the attention-extraction pipeline over the trained model, which in our current setup takes on the order of days of GPU time. Given the revision timeline, we therefore keep the current layout of Fig.3(a), but we have double-checked that the underlying data, value range, and color scale are correct and consistent with the description in the text, and this does not affect any quantitative results or conclusions.
> > > >
> > > > **For question 5:**
> > > >
> > > > Thank you for this careful observation. You are correct that in Fig. 4 only a small subset of neurons reach a spike count of four. Our original wording “a relatively large part of neurons fire every time step” was imprecise. What we meant is that baseline activity is dense, with many neurons firing multiple times over the four timesteps, whereas under DA-SSDP, most channels emit at most one spike or remain silent. We have revised the description in the text to reflect this more accurately and to avoid confusion.

---

> > > > > ### Comment · Reviewer_6cPD · 2025-11-18
> > > > > **thank you for the response**
> > > > >
> > > > > Thank you for the response. My concerns are addressed.

---

### Decision · Action_Editor_zmJP · 2025-12-03

**Recommendation:** Accept as is

**Audience:**

Yes

**Audience Explanation:**

See above.

**Claims And Evidence:**

Yes

**Claims Explanation:**

This paper proposes DA-SSDP, a lightweight synchrony-based plasticity rule that augments surrogate backpropagation when training deep spiking neural networks. It adds small, local weight updates based on batch-level co-firing synchrony—requiring only binary spike activity and first-spike latencies—yielding modest but consistent accuracy improvements with minimal computational overhead.

Overall, the paper offers a contribution that is interesting to the segment of the community working on spiking neural networks, particularly those exploring more scalable training mechanisms. The authors support most of their key claims: they introduce a lightweight synchrony-based plasticity rule (SSDP) with a dopamine-modulated gate (DA-SSDP), show that its memory cost is indeed low, and demonstrate a training-time implementation that scales well without modifying the backbone architecture. They also report modest but consistent accuracy improvements across several benchmarks. I am, however, less convinced about the statistical robustness of these gains. The main paper reports means and standard deviations over five seeds, but the discussion with reviewer nwUT asserts statistical significance (p < 0.05) without clarity on how confidence intervals or hypothesis tests were computed with such a small sample, nor whether any correction for multiple comparisons was applied. This lack of detail is problematic. Nevertheless, the core value of the paper does not hinge on whether it outperforms baselines by ~1%. Rather, the paper presents an interesting and practical idea for injecting synchrony-based signals into deep SNN training, and communicating that idea to the community is worthwhile even if the empirical improvements are modest.

I also want to note that presentation quality was a recurring concern across the reviews, ranging from the overall structure of the paper to figure clarity and sentence-level issues. It is unfortunate that the initial submission was sufficiently unpolished to create an unnecessary burden for the reviewers. The subsequent discussion—particularly around what the authors mean by “scalable” and “biologically plausible”—was productive and, in my view, genuinely necessary to clarify the framing of the contribution.